



# Towards a model for structured mass movements: the OpenLISEM Hazard model 2.0a

Bastian van den Bout*[1] Theo van Asch[2] Wei Hu[2] Chenxiao X. Tang[3] Olga Mavrouli[1] Victor G. Jetten[1] Cees J. van Westen[1]

[1]University of Twente, Faculty of Geo-Information Science and Earth Observation

[2]Chengdu university of Technology, State key Laboratory of Geohazard Preventaion and GeoEnvironment Protection

[3]Institute of Mountain Hazards and Environment, Chinese Academy of Sciences

*Correspondence to*: Bastian van den Bout (b.vandenbout@utwente.nl)

**Abstract**

Mass movements such as debris flows and landslide differ in behavior due to their material properties and internal forces. Models employ generalized multi-phase flow equations to adaptively describe these complex flow types. However, models commonly assume unstructured and fragmented flow after initiation of movement. In this work, existing work on two-phase mass movement equations are extended to include a full stress-strain relationship that allows for runout of (semi-) structured fluid-solid masses. The work provides both the three-dimensional equations and depth-averaged simplifications. The equations are implemented in a hybrid Material Point Method (MPM) which allows for efficient simulation of stress-strain relationships on discrete smooth particles. Using this framework, the developed model is compared to several flume experiments of clay blocks impacting fixed obstacles. Here, both final deposit patterns and fractures compare well to simulations. Additionally, numerical tests are performed to showcase the range of dynamical behavior produced by the model. Important processes such as fracturing, fragmentation and fluid release are captured by the model. While this provides an important step towards complete mass movement models, several new opportunities arise such as ground-water flow descriptions and application to fragmenting mass movements and block-slides.



**Introduction**

The earths rock cycle involves sudden release and gravity-driven transport of sloping materials. These mass movements have a significant global impact in financial damage and casualties (Nadim et al., 2006; Kjekstad & Highland, 2009). Understanding the physical principles at work at their initiation and runout phase allows for better mitigation and adaptation to the hazard they induce (Corominas et al., 2014). Many varieties of gravitationally-driven mass movements have been categorized according to their material physical parameters and type of movement. Examples are slides, flows and falls consisting of soil, rocks or debris (Varnes, 1987). Major factors in determining the dynamics of mass movement runout are the composition of the moving material and the forces during initiation and runout. Physically-based models attempt to describe the internal and external forces of all these mass movements in a generalized form (David & Richard, 2011; Pudasaini, 2012; Iverson & George, 2014). This allows these models to be applied to a wide variety of cases, while improving predictive range.

Dynamics of geophysical flows are complex and depend on a variety of forces due to their multi-phase interactions (Hutter et al., 1996). Generally, understanding and prediction of geophysical flows takes place through numerical modelling of the flow. A variety of both one, two and three- dimensional sets of equations exist to describe the advection and forces that determine the dynamics of geophysical flows. Examples that simulated a single mixed material (Rickenmann et al., 2006; O'Brien et al., 2007; Luna et al., 2012; van Asch et al., 2014). Two phase models describe both solids, fluids and their interactions and provide additional detail and generalize in important ways (Sheridan et al., 2005; Pitman & Le, 2005; Pudasaini, 2012; George & Iverson, 2014; Mergili et al., 2017). Recently, a three-phase model has been developed that includes the interactions between small and larger solid phases (Pudasaini & Mergili, 2019). Typically, implemented forces include gravitational forces and, depending on the rheology of the equations, drag forces, viscous internal forces and a plasticity-criterion.

A major assumption made for current models is the a fully mixed and fragmented nature of the material (Iverson & Denlinger 2001; Pudasaini & Hutter, 2003). This assumption is invalid for any structured mass movement. Observations of mass movement types indicate that mixing and fracturing is not a necessary process (Varnes, 1987). Instead, block or slide movement can retain structure during their dynamic stage, as the material is able to resists the internal deformation stresses. Some models do a non-Newtonian viscous yield stress based on depth-averaged strain estimations (Boetticher et al., 2016; Fornes et al., 2017; Pudasaini & Mergili, 2019). However, this approach lacks the process of fragmentation and internal failure. Thus, within current mass movement models, there might be improvements available from assuming non-fragmented movement. This would allow for description of structured mass movement dynamics.

The general importance of the initially structured nature of mass movement material is observed for a variety of reasons. First, block slides are an important subset of mass movement types (Hayir, 2003; Beutner et al., 2008; Tang et al., 2008). This type of mass movement features some cohesive structure to the dynamic material in the movement phase. Secondly, during movement, the spatial gradients in local acceleration induce strain and stress that results in fracturing. This process, often called fragmentation in relation to structured mass movements, can be of crucial importance for mass movement dynamics (Davies & McSaveney, 2009; Delaney & Evans, 2014; Dufresne et al., 2018; Corominas et al. 2019). Lubricating effect from basal fragmentation can enhance velocities and runout distance significantly (Davies et al., 2006; Tang et al., 2009). Otherwise, fragmentation generally influences the rheology of the movement by altering grain-grain interactions (Zhou et al., 2005). The importance of structured material dynamics is further indicated by engineering studies on rock behavior and fracture models (Kaklauskas & Ghaboussi, 2001; Ngekpe et al., 2016; Dhanmeher, 2017)

In this paper, existing two-phase generalized debris flow equations are adapted to describe runout of a arbitrarily structured two-phase Mohr-Coulomb material. The second section of this work provides the derivation of the extensive set of equations that describe structured mass movements in a generalized manner. The third section validates the developed model by comparison with results from controlled flume runout experiments. Additionally, this section shows numerical simulation examples that highlight fragmentation behavior and its influence on runout dynamics. Finally, in section four, a discussion on the potential usage of the presented model is provided together with reflection on important opportunities of improvement.

1. **A set of debris flow equations incorporating internal structure**

**1.1 Structured mass movements**

Initiation of gravitational mass flows occurs when sloping material is released. The instability of such materials is generally understood to take place along a failure plane (Zhang et al., 2011, Stead & Wolter, 2015). Along this plane, forces exerted due to gravity and possible seismic accelerations can act as a driving force towards the downslope direction, while a normal-force on the terrain induces a resisting force (Xie et al., 2006).





When internal stress exceeds a specified criteria, commonly described using Mohr-Coulomb theory, fracturing
occurs, and the material becomes dynamic. Observations indicate material can initially fracture predominantly at
the failure plane (Tang et al., 2009 Davies et al., 2006). Full finite-element modelling of stability confirms no
fragmentation occurs at initiation, and runout can start as a structured mass (Matsui & San, 1992; Griffiths &
Lane, 1999).

Once movement is initiated, the material is accelerated. Due to spatially non-homogeneous acceleration,
either caused by a non-homogeneous terrain slope, or impact with obstacles, internal stress can build within the
moving mass. The stress state can reach a point outside the yield surface, after which some form of deformation
occurs (e.g. Plastic, Brittle, ductile) (Loehnert et al., 2008). In the case of rock or soil material, elastic/plastic
deformation is limited and fracturing occurs at relatively low strain values (Kaklauskas & Ghaboussi, 2001;
Dhanmeher., 2017). Rocks and soil additionally show predominantly brittle fracturing, where strain increments
at maximum stress are small (Bieniawaski, 1967; Price, 2016; Husek et al., 2016). For soil matrices, cohesive
bonds between grains originate from causes such as cementing, frictionl contacts and root networks (Cohen et
al., 2009). Thus, the material breaks along either the grain-grain bonds or on the molecular level. In practice, this
processes of fragmentation has been both observed and studied frequently. Cracking models for solids use stress-
strain descriptions of continuum mechanics (Menin et al., 2009; Ngekpe et al., 2016). Fracture models frequently
use Smooth Particle Hydrodynamics (SPH) since a Lagrangian, meshfree solution benefits possible fracturing
behavior (Maurel & Combescure, 2008; Xu et al., 2010; Osorno & Steeb, 2017). Within the model developed
below, knowledge from fracture-simulating continuum mechanical models is combined with finite element fluid
dynamic models.
**1.2 Model description**

We define two phases, solids and fluids, within the flow, indicated by $s$ and $f$ respectively. A specified
fraction of solids within this mixture is at any point part of a structured matrix. This structured solid phase,
indicated by $sc$ envelops and confines a fraction of the fluids in the mixture, indicates as $fc$. The solids and
fluids are defined in terms of the physical properties such as densities ($\rho_f, \rho_s$) and volume fractions ($\alpha_f =$
$\frac{s}{f+s}, \alpha_s = \frac{f}{f+s}$). The confined fractions of their respective phases are indicated as $f_{sc}$ and $f_{fc}$ for the volume
fraction of confined solids and fluids respectively (Equations 1,2 and 3).
1.  $\alpha_s + \alpha_f = 1$
2.  $\alpha_s\left(f_{sc} + (1 - f_{sc})\right) + \alpha_f\left(f_{fc} + (1 - f_{fc})\right) = 1$
3.  $\left(f_{sc} + (1 - f_{sc})\right) = \left(f_{fc} + (1 - f_{fc})\right) = 1$

For the solids, additionally internal friction angle ($\phi_s$) and effective (volume-averaged) material size
($d_s$) are defined. We additionally define $\alpha_c = \alpha_s + f_{fc}\alpha_f$ and $\alpha_u = (1 - f_{fc})\alpha_f$ to indicate the solids with
confined fluids and free fluid phases respectively. These phases have a volume-averaged density $\rho_{sc}, \rho_f$. We let
the velocities of the unconfined fluid phase ($\alpha_u == (1 - f_{fc})\alpha_f$) be defined as $u_u = (u_u, v_u)$. We assume
velocities of the confined phases ($\alpha_c = \alpha_s + f_{fc}\alpha_f$) can validly be assumed to be identical to the velocities of
the solid phase, $u_c = (u_c, v_c) = u_s = (u_s, v_s)$. A schematic depiction of the represented phases is shown in
Figure 1.





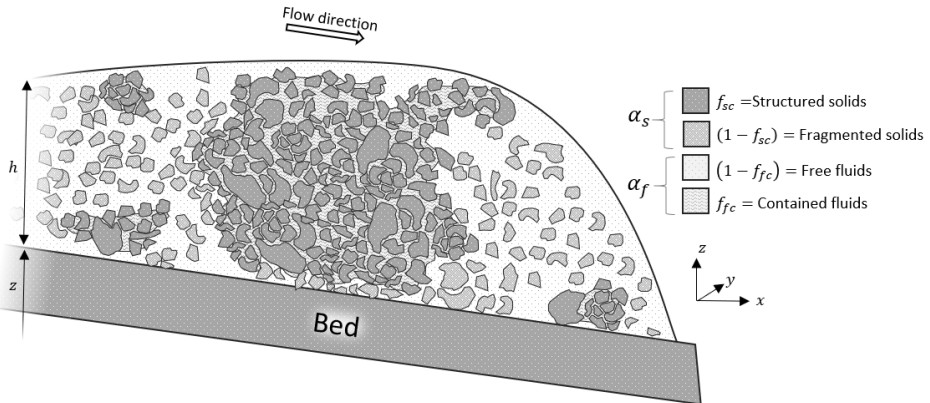


Figure 1 A schematic depiction of the flow contents. Both structured and unstructured solids are present. Fluids can be either free, or confined by the structured solids.


A major assumption is made here concerning the velocities of both the confined and free solids (sc and
s), that have a shared averaged velocity ($u_s$). We deliberately limit the flow description to two phases, opposed
to the innovative work of Pudasaini & Mergili (2019) that develop a multi-mechanical three-phase model. This
choice is motivated by considerations of applicability (reducing the number of required parameters), the infancy
of three-phase flow descriptions and finally the general observations of the validity of this assumption (Ishii,
1975; Ishii & Zuber, 1979; Drew, 1983; Jakob et al, 2005; George & Iverson, 2016).
The movement of the flow is described initially by means of mass and momentum conservation
(Equations 4 and 5).
4.  $\frac{\partial \alpha_c}{\partial t} + \nabla \cdot (\alpha_c \boldsymbol{u}_c) = 0$
5.  $\frac{\partial \alpha_u}{\partial t} + \nabla \cdot (\alpha_u \boldsymbol{u}_u) = 0$
Here we add the individual forces based on the work of Pudasaini & Hutter (2003), Pitman & Le
(2005), Pudasaini (2012), Pudasaini & Fischer (2016) and Pudasaini & Mergili (2019) (Equations 6 and 7).
6.  $\frac{\partial}{\partial t}(\alpha_c \rho_c \boldsymbol{u}_c) + \nabla \cdot (\alpha_c \rho_c \boldsymbol{u}_c \otimes \boldsymbol{u}_c) = \alpha_c \rho_c \boldsymbol{f} - \nabla \cdot \alpha_c \boldsymbol{T}_c + p_c \nabla \alpha_c + \boldsymbol{M}_{DG} + \boldsymbol{M}_{vm}$
7.  $\frac{\partial}{\partial t}(\alpha_u \rho_f \boldsymbol{u}_u) + \nabla \cdot (\alpha_u \rho_f \boldsymbol{u}_u \otimes \boldsymbol{u}_u) = \alpha_u \rho_f \boldsymbol{f} - \nabla \cdot \alpha_u \boldsymbol{T}_u + p_f \nabla \alpha_u - \boldsymbol{M}_{DG} - \boldsymbol{M}_{vm}$
Where $\boldsymbol{f}$ is the body force (among which is gravity), $\boldsymbol{M}_{DG}$ is the drag force, $\boldsymbol{M}_{vm}$ is the virtual mass
force and $\boldsymbol{T}_c, \boldsymbol{T}_u$ are the stress tensors for solids with confined fluids and unconfined phases respectively. The
virtual mass force described the additional work required by differential acceleration of the phases. The drag
force describes the drag along the interfacial boundary of fluids and solids. The body force describes external
forces such as gravitational acceleration and boundary forces. Finally, the stress tensors describe the internal
forces arising from strain and viscous processes. Both the confined and unconfined phases in the mixture are
subject to stress tensors ($T_c$, and $T_u$), for which the gradient acts as a momentum source. Additionally, we follow
Pudasaini (2012) and add a buoyancy force ($p_c \nabla \alpha_c$ and $p_f \nabla \alpha_u$).
**Stress Tensors, Describing internal structure**
Based on known two-phase mixture theory, the internal and external forces acting on the moving
material are now set up. This results in several unknowns such as the stress tensors ($\boldsymbol{T}_c$ and $\boldsymbol{T}_u$, described by the
constitutive equation), the body force ($\boldsymbol{f}$), the drag force ($\boldsymbol{M}_{DG}$) and the virtual mass force ($\boldsymbol{M}_{VM}$). This section
will first describe the derivation of the stress tensors. These describe the internal stress and viscous effects. To
describe structured movements, these require a full stress-strain relationship which is not present in earlier
generalized mass movements model. Afterwards, existing derivation of the body, drag and virtual mass force are
altered to conform the new constitutive equation.
Our first step in defining the momentum source terms in equations 6 and 7 is the definition of the fluid
and solid stress tensors. Current models typically follow the assumptions made by Pitman & Le (2005), who
indicate: "*Proportionality and alignment of the tangential and normal forces are imposed as a basal boundary*





*condition is assumed to hold throughout the layer of flowing material ... following Rankine (1857) and Terzaghi*
*(1936), an earth pressure relation is assumed for diagonal stress components*". Here, the earth pressure
relationship is a vertically-averaged analytical solution for lateral forces exerted by an earth wall. Thus,
unstructured columns of moving mixtures are assumed. Here, we aim to use the full Mohr-Coulomb relations.
Describing the internal tress of soil and rock matrices is commonly achieved be elastic-plastic simulations of the
materials stress-strain relationship. Since we aim to model a full stress description, the stress tensor is equal to
the elasto-plastic stress tensor (Equation 8).

8.   $T_c = \boldsymbol{\sigma}$

Where $\boldsymbol{\sigma}$ is the elasto-plastic stress tensor for solids. The stress can be divided into the deviatoric and
non-deviatoric contributions (Equation 9). The non-deviatoric part acts normal on any plane element (in the
manner in which a hydrostatic pressure acts equal in all directions). Note that we switch to tensor notation when
describing the stress-strain relationship. Thus, superscripts ($\alpha$ and $\beta$) represent the indices of basis vectors (x, y
or z axis in Euclidian space), and obtain tensor elements. Additionally, the Einstein convention is followed
(automatic summation of non-defined repeated indices in a single term).
9.   $\sigma^{\alpha\beta} = s^{\alpha\beta} + \frac{1}{3}\sigma^{\gamma\gamma}\delta^{\alpha\beta}$
Where $s$ is the deviatoric stress tensor and $\delta^{\alpha\beta} = [\alpha = \beta]$ is the Kronecker delta.
Here, we define the elasto-plastic stress ($\sigma$) based on a generalized Hooke-type law in tensor notation
(Equation 10 and 11) where plastic strain occurs when the stress state reaches the yield criterion (Spencer, 2004;
Necas & Hiavecek, 2007; Bui et al., 2008).
10.   $\dot{\epsilon}_{elastic}^{\alpha\beta} = \frac{\dot{s}^{\alpha\beta}}{2G} + \frac{1-2\nu}{E}\dot{\sigma}^m\delta^{\alpha\beta}$
11.   $\dot{\epsilon}_{plastic}^{\alpha\beta} = \dot{\lambda}\frac{\partial g}{\partial \sigma^{\alpha\beta}}$
Where $\dot{\epsilon}_{elastic}$ is the elastic strain tensor, $\dot{\epsilon}_{plastic}$ is the plastic strain tensor, $\dot{\sigma}^m$ is the mean stress rate
tensor, $\nu$ is Poisson's ratio, $E$ is the elastic Young's Modulus, $G$ is the shear modulus, $\dot{s}$ is the deviatoric shear
stress rate tensor, $\dot{\lambda}$ is the plastic multiplier rate and $g$ is the plastic potential function. Additionally, the strain
rate is defined from velocity gradients as equation 12.
12.   $\dot{\epsilon}_{total}^{\alpha\beta} = \dot{\epsilon}_{elastic}^{\alpha\beta} + \dot{\epsilon}_{plastic}^{\alpha\beta} = \frac{1}{2}\left(\frac{\partial u_c^\alpha}{\partial x^\beta} - \frac{\partial u_c^\beta}{\partial x^\alpha}\right)$
By solving equations 9, 10 and 11 for $\dot{\sigma}$, a stress-strain relationship can be obtained (Equation 13) (Bui
et al., 2008).
13.   $\dot{\sigma}^{\alpha\beta} = 2G\dot{e}^{\gamma\gamma}\delta^{\alpha\beta} + K\dot{\epsilon}^{\gamma\gamma}\delta^{\alpha\beta} - \dot{\lambda}\left[\left(K - \frac{2G}{3}\right)\frac{\partial g}{\partial \sigma^{mn}}\delta^{mn}\delta^{\alpha\beta} + 2G\frac{\partial g}{\partial \sigma^{\alpha\beta}}\right]$
Where $\dot{e}$ is the deviatoric strain rate ($\dot{e}^{\alpha\beta} = \dot{\epsilon}^{\gamma\gamma} - \frac{1}{3}\dot{\epsilon}^{\alpha\beta}\delta^{\alpha\beta}$), $\psi$ is the dilatancy angle and K is the
elastic bulk modulus and the material parameters defined from from $E$ and $\nu$ (Equation 14).
14.   $K = \frac{E}{3(1-2\nu)}, \;\; G = \frac{E}{2(1+\nu)}$
Fracturing or failure occurs when the stress state reaches the yield surface, after which plastic
deformation occurs. The rate of change of the plastic multiplier specifies the magnitude of plastic loading and
must ensure a new stress state conforms to the conditions of the yield criterion. By means of substituting
equation 13 in the consistency condition ($\frac{\partial f}{\partial \sigma^{\alpha\beta}}d\sigma^{\alpha\beta} = 0$), the plastic multiplier rate can be defined (Equation
15) (Bui et al., 2008).
15.   $\dot{\lambda} = \frac{2G\epsilon^{\alpha\beta}\frac{\partial f}{\partial \sigma^{\alpha\beta}} + \left(K - \frac{2G}{3}\right)\dot{\epsilon}^{\gamma\gamma}\frac{\partial f}{\partial \sigma^{\alpha\beta}}\sigma^{\alpha\beta}\delta^{\alpha\beta}}{2G\frac{\partial f}{\partial \sigma^{mn}}\frac{\partial g}{\partial \sigma^{mn}} + \left(K - \frac{2G}{3}\right)\frac{\partial f}{\partial \sigma^{mn}}\delta^{mn}\frac{\partial g}{\partial \sigma^{mn}}\delta^{mn}}$
The yield criteria specifies a surface in the stress-state space that the stress state can not pass, and at
which plastic deformation occurs. A variety of yield criteria exist, such as Mohr-Coulomb, Von Mises, Ducker-
Prager and Tresca (Spencer, 2004). Here, we employ the Ducker-Prager model fitted to Mohr-Coulomb material
parameters for its accuracy in simulating rock and soil behavior, and numerical stability (Spencer, 2004; Bui et
al., 2008) (Equation 16 and 17).
16.   $f(I_1, J_2) = \sqrt{J_2} + \alpha_\phi I_1 - k_c = 0$
17.   $g(I_1, J_2) = \sqrt{J_2} + \alpha_\phi I_1 \sin(\psi)$



Where $I_1$ and $J_2$ are tensor invariants (Equation 18 and 19).
18. $I_1 = \sigma^{xx} + \sigma^{yy} + \sigma^{zz}$
19. $J_2 = \frac{1}{2} s^{\alpha\beta} s^{\alpha\beta}$
Where the Mohr-Coulomb material parameters are used to estimate the Ducker-Prager parameters
(Equation 20).
20. $\alpha_\phi = \frac{\tan(\phi)}{\sqrt{9 + 12\tan^2\phi}}, \quad k_c = \frac{3c}{\sqrt{9 + 12\tan^2\phi}}$
Using the definitions of the yield surface and stress-strain relationship, combining equations 13, 15, 16
and 17, the relationship for the stress rate can be obtained (Equation 21 and 22).
21. $\dot{\sigma} = 2G\dot{e}^{\alpha\beta} + K\dot{\epsilon}^{\gamma\gamma}\delta^{\alpha\beta} - \dot{\lambda}\left[9K\sin\psi\,\delta^{\alpha\beta} + \frac{G}{\sqrt{J_2}}s^{\alpha\beta}\right]$
22. $\dot{\lambda} = \frac{3\alpha K\dot{\epsilon}^{\gamma\gamma} + \left(\frac{G}{\sqrt{J_2}}\right)s^{\alpha\beta}\dot{e}^{\alpha\beta}}{27\alpha_\phi K\sin\psi + G}$
In order to allow for the description of large deformation, the Joumann stress rate can be used, which is
a stress-rate that is independent from a frame of reference (Equation 23).
23. $\hat{\sigma} = \sigma^{\alpha\gamma}\dot{\omega}^{\beta\gamma} + \sigma^{\gamma\beta}\dot{\omega}^{\alpha\gamma} + 2G\dot{e}^{\alpha\beta} + K\dot{\epsilon}^{\gamma\gamma}\delta^{\alpha\beta} - \dot{\lambda}\left[9K\sin\psi\,\delta^{\alpha\beta} + \frac{G}{\sqrt{J_2}}s^{\alpha\beta}\right]$
Where $\dot{\omega}$ is the spin rate tensor, as defined by equation 24.
24. $\dot{\omega}^{\alpha\beta} = \frac{1}{2}\left(\frac{\partial v^\alpha}{\partial x^\beta} - \frac{\partial v^\beta}{\partial x^\alpha}\right)$
Due to the strain within the confined material, the density of the confined solid phase ($\rho_c$) evolves
dynamically according to equation 25.
25. $\rho_c = f_{sc}\rho_s\frac{\epsilon_{v0}}{\epsilon_v} + (1 - f_{sc})\rho_s + f_{fc}\rho_f$
Where $\epsilon_v$ is the total volume strain, $\epsilon_v \approx \epsilon_1 + \epsilon_2 + \epsilon_3$, $\epsilon_i$ is one of the principal components of the
strain tensor. Since we aim to simulate brittle materials, where volume strain remains relatively low, we assume
that changes in density are small compared to the original density of the material ($\frac{\partial \rho_c}{\partial t} \ll \rho_c$).
**Fragmentation**
Brittle fracturing is a processes commonly understood to take place once a material internal stress has
reached the yield surface, and plastic deformation has been sufficient to pass the ultimate strength point (Maurel
& Cumescure, 2008; Husek et al., 2016). A variety of approaches to fracturing exist within the literature (Ma et
al., 2014; Osomo & Steeb, 2017). FEM models use strain-based approaches (Loehnert et al., 2008). For SPH
implementations, as will be presented in this work, distance-based approaches have provided good results
(Maurel & Cumbescure, 2008). Other works have used strain-based fracture criteria (Xu et al., 2010) .
Additionally, dynamic degradation of strength parameters have been implemented (Grady & Kipp, 1980; Vuyst
& Vignjevic, 2013; Williams, 2019). Comparisons with observed fracture behavior has indicated the predictive
value of these schemes (Xu et al., 2010; Husek et al., 2016). We combine the various approaches to best fit the
dynamical multi-phase mass movement model that is developed. Following, Grady & Kipp (1980) and we
simulate a degradation of strength parameters. Our material consists of a soil and rock matrix. We assume
fracturing occurs along the inter-granular or inter-rock contacts and bonds (see also Cohen et al., 2009). Thus,
cohesive strength is lost for any fractured contacts. We simulate degradation of cohesive strength according to a
volume strain criteria. When the stress state lies on the yield surface (the set of critical stress states within the 6-
dimensional stress-space), during plastic deformation, strain is assumed to attribute towards fracturing. A critical
volume strain is taken as material property, and the breaking of cohesive bonds occurs based on the relative
volume strain. Following Grady & Kipp (1980) and Vuyst & Vignjevic (2013), we assume that the degradation
behavior of the strength parameter is distributed according to a probability density distribution. Commonly, a
Weibull-distribution is used (Williams, 2019). Here, for simplicity, we use a uniform distribution of cohesive
strength between 0 and $2c_0$, although any other distribution can be substituted. Thus, the expression governing
cohesive strength becomes equation 26
26. $\frac{\partial c}{\partial t} = \begin{cases} -c_0 \frac{1}{2}\frac{\left(\frac{\epsilon_v}{\epsilon_{v0}}\right)}{\epsilon_c} & f(I_1, J_2) \geq 0, c > 0 \\ 0 & otherwise \end{cases}$





Where $c_0$ is the initial cohesive strength of the material, $\epsilon_{v0}$ is the initial volume, $\left(\frac{\epsilon_v}{\epsilon_{v0}}\right)$ is the fractional
volumetric strain rate, $\epsilon_c$ is the critical fractional volume strain for fracturing.

**Water partitioning**

During the movement of the mixed mass, the solids can thus be present as a structured matrix. Within
such a matrix, a fluid volume can be contained (e.g. as originating from a ground water content in the original
landslide material). These fluids are typically described as groundwater flow following Darcy's law, which poses
a linear relationship between pressure gradients and flow velocity through a soil matrix. In our case, we assumed
the relative velocity of water flow within the granular solid matrix as very small compared to both solid
velocities and the velocities of the free fluids. As an initial condition of the material, some fraction of the water
is contained within the soil matrix ($f_{fc}$). Additionally, for loss of cohesive structure within the solid phase, we
transfer the related fraction of fluids contained within that solid structure to the free fluids.
27. $\frac{\partial f_{fc}}{\partial t} = -\frac{\partial(1-f_{fc})}{\partial t} = \begin{cases} -f_{fc}\frac{c_0}{c}\frac{\max(0.0,\epsilon_v)}{\epsilon_f} & f(I_1,J_2) \geq 0, c > 0 \\ 0 & otherwise \end{cases}$
28. $\frac{\partial f_{sc}}{\partial t} = -\frac{\partial(1-f_{sc})}{\partial t} = \begin{cases} -f_{sc}\frac{c_0}{c}\frac{\max(0.0,\epsilon_v)}{\epsilon_f} & f(I_1,J_2) \geq 0, c > 0 \\ 0 & otherwise \end{cases}$
Beyond changes in $f_{fc}$ through fracturing of structured solid materials, no dynamics are simulated for
in- or outflux of fluids from the solid-matrix. The initial volume fraction of fluids in the solid matrix defined by
($f f_{fc}$ and $s f_{sc}$) remains constant throughout the simulation. The validity of this assumption can be based on the
slow typical fluid velocities in a solid matrix relative to fragmented mixed fluid-solid flow velocities (Kern,
1995; Saxton and Rawls, 2006). While the addition of evolving saturation would extend validity of the model, it
would require implementation of pretransfer-functions for evolving material properties, which is beyond the
scope of this work. An important note on the points made above is the manner in which fluids are re-partitioned
after fragmentation. All fluids in fragmented solids are released, but this does not equate to free movement of the
fluids or a disconnection from the solids that confined them. Instead, the equations continue to connect the solids
and fluids through drag, viscous and virtual mass forces. Finally, the density of the fragmented solids is assumed
to be the initially set solid density. Any strain-induced density changes are assumed small relative to the initial
solid density ($\frac{\rho_c}{\rho_s} \ll 1$).

**Fluid Stresses**

The fluid stress tensor is determined by the pressure and the viscous terms (Equations 29 and 30).
Confined solids are assumed to be saturated and constant during the flow.
29. $\boldsymbol{T}_u = P_f \boldsymbol{I} + \boldsymbol{\tau}_f$
30. $\boldsymbol{\tau}_f = \eta_f[\nabla\boldsymbol{u}_u + (\nabla\boldsymbol{u}_c)^t] - \frac{\eta_f}{\alpha_u}\mathcal{A}(\alpha_u)(\nabla\alpha_c(\boldsymbol{u}_u - \boldsymbol{u}_c) + (\boldsymbol{u}_c - \boldsymbol{u}_u)\nabla\alpha_c)$
Where $\boldsymbol{I}$ is the identity tensor, $\boldsymbol{\tau}_f$ is the viscous stress tensor for fluids , $P_f$ is the fluid pressure, $\eta_f$ is the
dynamic viscosity of the fluids and $\mathcal{A}$ is the mobility of the fluids at the interface with the solids that acts as a
phenomenological parameter (Pudasaini, 2012).
The fluid pressure acts only on the free fluids here, as the confined fluids are moved together with the
solids. In equation 30, the second term is related to the non-Newtonian viscous force induced by gradients in
solid concentration. The effect as described by Pudasaini (2012) is induced by a solid-concentration gradient. In
case of unconfined fluids and unstructured solids ($f_{sf} = 1, f_{sf} = 1$). Within our flow description, we see no
direct reason to eliminate or alter this force with a variation in the fraction of confined fluids or structured solids.
We do only consider the interface between solids and free fluids as an agent that induces this effect, and
therefore the gradient of the gradient of the solids and confined fluids ($\nabla(\alpha_s + f_{fc}\alpha_f) = \nabla\alpha_c$) is used instead of
the total solid phase ($\nabla\alpha_s$).

**Drag force and Virtual Mass**

Our description of the drag force follows the work of Pudasaini (2012) and Pudasaini (2018), where a
generalized two-phase drag model is introduced and enhanced. We split their work into a contribution from the
fraction of structured solids ($f_{sc}$) and unconfined fluids ($1 - f_{fc}$) (Equation 31).
31. $\mathcal{C}_{DG} = \frac{f_{sc}\alpha_c\alpha_u(\rho_c-\rho_f)g}{U_{T,c}(\mathcal{G}(Re))+S_p}(\boldsymbol{u}_u - \boldsymbol{u}_c)|\boldsymbol{u}_u - \boldsymbol{u}_c|^{j-1} + \frac{(1-f_{sc})\alpha_c\alpha_u(\rho_s-\rho_f)g}{U_{T,uc}(\mathcal{PF}(Re_p)+(1-\mathcal{P})\mathcal{G}(Re))+S_p}(\boldsymbol{u}_u - \boldsymbol{u}_c)|\boldsymbol{u}_u - \boldsymbol{u}_c|^{j-1}$



Where $U_{T,c}$ is the terminal or settling velocity of the structures solids, $U_{T,uc}$ is the terminal velocity of
the unconfined solids, $\mathcal{P}$ is a factor that combines solid- and fluid like contributions to the drag force, $\mathcal{G}$ is the
solid-like drag contribution, $\mathcal{F}$ is the fluid-like drag contribution and $S_p$ is the smoothing function (Equation 32
and 34). The exponent $j$ indicates the type of drag: linear ($j = 0$) or quadratic ($j = 1$).
Within the drag, the following functions are defined:
32. $F = \frac{\gamma}{180}\left(\frac{\alpha_f}{\alpha_s}\right)^3 Re_P, \ G = \alpha_f^{M(Re_P)-1}$
33. $S_p = \left(\frac{\mathcal{P}}{\alpha_c} + \frac{1-\mathcal{P}}{\alpha_u}\right)\mathcal{K}$
34. $\mathcal{K} = |\alpha_c \boldsymbol{u}_c + \alpha_u \boldsymbol{u}_u| \approx 10 \ ms^{-1}$
Where $M$ is a parameter that varies between 2.4 and 4.65 based on the Reynolds number (Pitman & Le,
2005). The factor $\mathcal{P}$ that combines solid-and fluid like contributions to the drag, is dependent on the volumetric
solid content in the unconfined and unstructured materials ($\mathcal{P} = \left(\frac{\alpha_s(1-f_{sc})}{\alpha_f(1-f_{fc})}\right)^m$ with $m \approx 1$. Additionally we
assume the factor $\mathcal{P}$, is zero for drag originating from the structured solids. As stated by Pudasaini & Mergili
(2019) "As limiting cases: $\mathcal{P}$ suitably models solid particles moving through a fluid". In our model, the drag
force acts on the unconfined fluid momentum ($u_{uc}\alpha_f(1 - f_{fc})$). For interactions between unconfined fluids and
structured solids, larger blocks of solid structures are moving through fluids that contains solids of smaller size.
Virtual mass is similarly implemented based on the work of Pudasaini (2012) and Pudasaini & Mergili
(2019) (Equation 35). The adapted implementation considers the solids together with confined fluids to move
through a free fluid phase.
35. $\mathcal{C}_{VMG} = \alpha_c \rho_u \left(\frac{1}{2}\left(\frac{1+2\alpha_c}{\alpha_u}\right)\right)\left(\left(\frac{\partial u_u}{\partial t} + u_u \cdot \nabla u_u\right) - \left(\frac{\partial u_c}{\partial t} + u_c \cdot \nabla u_c\right)\right)$
Where $C_{DG} = \frac{1}{2}\left(\frac{1+2\alpha_c}{\alpha_u}\right)$ is the drag coefficient.

**310   boundary conditions**

Finally, following the work of Iverson & Denlinger (2001), Pitman & Le (2005) and Pudasaini (2012), a
boundary condition is applied to the surface elements that contact the flow (Equation 36).
36. $|\boldsymbol{S}| = Ntan(\phi)$
Where $N$ is the normal pressure on the surface element and $\boldsymbol{S}$ is the shear stress.

**315   1.3   Depth-Averaging**

The majority of the depth-averaging in this works is analogous to the work of Pitman & Le (2005),
Pudasaini (2012) and Pudasini & Mergili (2019). Depth-averaging through integration over the vertical extent of
the flow can be done based on several useful and often-used assumptions: $\frac{1}{h}\int_0^h x \, dh = \bar{x}$, for the velocities ($u_u$
and $u_c$), solid, fluid and confined fractions ($\alpha_f$, $\alpha_s$, $f_{fc}$ and $f_{sc}$) and material properties ($\rho_u$, $\phi$ and $c$). Besides
these similarities and an identical derivation of depth-averaged continuity equations, three major differences
arise.
**i)Fluid pressure**
Previous implementations of generalized two-phase debris flow equations have commonly assumed hydrostatic
pressure ($\frac{\partial p}{\partial z} = g^z$) (Pitman & Le, 2005; Pudasaini, 2012; Abe & Konagai, 2016). Here we follow this
assumption for the fluid pressure at the base and solid pressure for unstructured material (Equations 37 and 38 ).
37. $P_{b_{s,u}} = -(1 - \gamma)\alpha_s g^z h$
38. $P_{b_u} = -g^z h$
Where $\gamma = \frac{\rho_f}{\rho_s}$ is the density ratio (not to be confused with a tensor index when used in superscript) (-).
However, larger blocks of structure material can have contact with the basal topography. Due to density
differences, larger blocks of solid structures are likely to move along the base (Pailhia & Pouliquen, 2009;
George & Iverson, 2014). If these blocks are saturated, water pressure propagates through the solid matrix and
hydrostatic pressure is retained. However, in cases of an unsaturated solid matrix that connects to the base,
hydrostatic pressure is not present there. We introduce a basal fluid pressure propagation factor $\mathcal{B}(\theta_{eff}, \overline{d_{sc}},..)$
which describes the fraction of fluid pressure propagated through a solid matrix (with $\theta_{eff}$ the effective



saturation, $\overline{d_{sc}}$ the average size of structured solid matrix blocks). This results in a basal pressure equal to
equation 39.
39. $P_{b_c} = -(1-f_{sc})(1-\gamma)\frac{(1-f_{sc})\alpha_s}{(1-f_{fc})\alpha_f} g^z h - f_{sc}(1-\gamma)\mathcal{B}\frac{(f_{sc})\alpha_s}{(f_{fc})\alpha_f} g^z h$
The basal pressure propagation factor ($\mathcal{B}$) should theoretically depend, similarly to the pedotransfer
function, mostly on saturation level, as a full saturation means perfect propagation of pressure through the
mixture, and low saturation equates to minimal pressure propagation (Saxton and Rawls., 2006). Additionally it
should depend on pedotransfer functions, and the size distribution of structured solid matrices within the
mixture. For low-saturation levels, it can be assumed no fluid pressure is retained. Combined with an assumed
soil matrix height identical to the total mixture height, this results in $\mathcal{B} = 0$. Assuming saturation of structures
solids results in a full propagation of pressures and $\mathcal{B} = 1$.
**ii)Stress-Strain relationship**
Depth-averaging the stress-strain relationship in equations 22 and 23 requires a vertical solution for the
internal stress. First, we assume any non-normal vertical terms are zero (Equation 40). Commonly, Rankines
earth pressure coefficients are used to express the lateral earth pressure by assuming vertical stress to be induced
by the basal solid pressure (Equation 41 and 42) (Pitman & Le, 2005; Pudasaini, 2012; Abe & Konagai, 2016).
40. $\sigma^{zx} = \sigma^{zy} = \sigma^{yz} = \sigma^{xz} = 0$
41. $\overline{\sigma^{zz}} = \frac{1}{2}P_{b_s}, \sigma^{zz}|_b = P_{b_s}$
42. $K_a = \frac{1-\sin(\phi)}{1+\sin(\phi)}, \quad K_p = \frac{1-\sin(\phi)}{1+\sin(\phi)}$
Here we enhance this with Bell's extension for cohesive soils (Equation 45) (Richard et al., 2017). This
lateral normal-directed stress term is added to the full stress-strain solution.
43. $\overline{\sigma_{xx}} = K\sigma_{zz}|_b - 2c\sqrt{K} + \frac{1}{h}\int_0^h \sigma_{xx}\, dh$
Finally, the gradient in pressure of the lateral interfaces between the mixture is added as a depth-
averaged acceleration term (Equation 44).
44. $S_{x_c} = \alpha_c(\frac{1}{h}\left(\frac{\partial(h\sigma^{xx})}{\partial x} + \frac{\partial(h\sigma^{yx})}{\partial y}\right)) + \cdots$
**iii)Depth-averaging other terms**
While the majority of terms allow for depth-averaging as proposed by Pudasaini (2012), an exception
arises. Depth-averaging of the vertical viscosity terms is required. The non-Newtonian viscous terms for the fluid
phase were derived assuming a vertical profile in the volumetric solid phase content. Here, we alter the
derivation to use this assumption only for the non-structured solids, as opposed to the structured solids where
$\frac{\partial \alpha_s}{\partial z} = 0$.
45. $\int_b^s \frac{\partial}{\partial z}\left(\frac{\partial \alpha_s}{\partial z}(u_u - u_c)\right) dz = \left[\frac{\partial \alpha_s}{\partial z}(u_u - u_c)\right]_b^s = (\overline{u_u} - \overline{u_c})\left[\frac{\partial \alpha_s}{\partial z}\right]_b^s = (\overline{u_u} - \overline{u_c})\left[\frac{\partial \alpha_s}{\partial z}\right]_b^s =$
$\frac{(\overline{u_u}-\overline{u_c})(1-f_{sc})\zeta\, \overline{\alpha_s}}{h}$
Where $\zeta$ is the shape factor for the vertical distribution of solids (Pudasaini, 2012). Additionally, the
momentum balance of Pudasaini (2012) ignores any deviatoric stress ($\tau_{xy} = 0$), following Savage and Hutter
(2007), and Pudasaini and Hutter (2007). Earlier this term was included by Iverson and Denlinger (2001), Pitman
and Le (2005) and Abe &Kanogai (2016). Here we include these terms since a full stress-strain relationship is
included.
**Basal frictions**
Additionally we add the Darcy-Weisbach friction, which is a Chezy-type friction law for the fluid phase
that provides drag (Delestre et al., 2014). This ensures that, without solid phase, a clear fluid does lose
momentum due to friction from basal shear. This was successfully done in Bout et al. (2018) and was similarly
assumed in Pudasaini and Fischer (2016) for fluid basal shear stress.
46. $S_f = \frac{g}{n^2}\frac{\mathbf{u_u}|\mathbf{u_u}|}{h^{\frac{4}{3}}}$
Where $n$ is Manning's surface roughness coefficient.
**Depth-averaged equations**





The following set of equations is thus finally achieved for depth-averaged flow over sloping terrain (Equations 47-71).

47. $\frac{\partial h}{\partial t} + \frac{\partial}{\partial x}[h(\alpha_u u_u + \alpha_c u_c)] + \frac{\partial}{\partial y}[h(\alpha_u u_u + \alpha_c u_c)] = R - I$

48. $\frac{\partial \alpha_c h}{\partial t} + \frac{\partial \alpha_c h u_c}{\partial x} + \frac{\partial \alpha_c h v_c}{\partial y} = 0$

49. $\frac{\partial \alpha_u h}{\partial t} + \frac{\partial \alpha_u h u_u}{\partial x} + \frac{\partial \alpha_u h v_u}{\partial y} = R - I$

50. $\frac{\partial}{\partial t}[\alpha_c h(u_c - \gamma_c C_{VM}(u_u - u_c))] + \frac{\partial}{\partial x}[\alpha_c h(u_c^2 - \gamma_c C_{VM}(u_u^2 - u_c^2))] + \frac{\partial}{\partial y}[\alpha_c h(u_c v_c - \gamma C(u_u v_u - u_c v_c))] = h S_{x_c}$

51. $\frac{\partial}{\partial t}[\alpha_c h(v_c - \gamma_c C_{VM}(v_u - v_c))] + \frac{\partial}{\partial x}[\alpha_c h(u_s v_s - \gamma_c C_{VM}(u_u v_u - u_c v_c))] + \frac{\partial}{\partial y}[\alpha_c h(v_c^2 - \gamma C_{VM}(v_u^2 - v_c^2))] = h S_{y_c}$

52. $\frac{\partial}{\partial t}[\alpha_u h(u_u - \frac{\alpha_c}{\alpha_u} C_{VM}(u_u - u_c))] + \frac{\partial}{\partial x}[\alpha_u h(u_u^2 - \frac{\alpha_c}{\alpha_u} C_{VM}(u_u^2 - u_c^2) + \frac{\beta_{x_u} h}{2})] + \frac{\partial}{\partial y}[\alpha_u h(u_u v_u - \gamma_c C_{VM}(u_u v_u - u_c v_c))] = h S_{x_u} - I u_u$

53. $\frac{\partial}{\partial t}[\alpha_u h(v_u - \frac{\alpha_c}{\alpha_u} C_{VM}(v_u - v_c))] + \frac{\partial}{\partial x}[\alpha_u h(u_u v_u - \frac{\alpha_c}{\alpha_u} C_{VM}(u_u v_u - u_c v_c))] + \frac{\partial}{\partial y}[\alpha_u h(v_u^2 - \gamma_c C_{Vm}(v_u^2 - v_c^2) + \frac{\beta_{y_u} h}{2})] = h S_{y_u} - I v_u$

54. $S_{x_c} = \alpha_c[g^x + \frac{1}{h}(\frac{\partial(h\sigma^{xx})}{\partial x} + \frac{\partial(h\sigma^{yx})}{\partial y}) - P_{b_c}(\frac{u_c}{|\vec{u_c}|}\tan\phi + \epsilon\frac{\partial b}{\partial x}) - \epsilon\alpha_c\gamma_c p_{b_u}[\frac{\partial h}{\partial x} + \frac{\partial b}{\partial x}] + C_{DG}(u_u - u_c)|\boldsymbol{u}_u - \boldsymbol{u}_c|^{J-1}$

55. $S_{y_c} = \alpha_c[g^y + \frac{1}{h}(\frac{\partial(h\sigma^{xy})}{\partial x} + \frac{\partial(h\sigma^{yy})}{\partial y}) - P_{b_c}(\frac{v_s}{|\vec{u_s}|}\tan\phi + \epsilon\frac{\partial b}{\partial y}) - \epsilon\alpha_c\gamma_c p_{b_u}[\frac{\partial h}{\partial y} + \frac{\partial b}{\partial y}] + C_{DG}(v_u - v_c)|\boldsymbol{v}_u - \boldsymbol{v}_c|^{J-1}$

56. $S_{x_u} = \alpha_u[g^x - \frac{\frac{1}{2}P_{b_u} h}{\alpha_u}\frac{\partial \alpha_c}{\partial x} + P_{b_u}\frac{\partial b}{\partial x} - \frac{\mathcal{A}\eta_u}{\alpha_u}(2\frac{\partial^2 u_u}{\partial x^2} + \frac{\partial^2 v_u}{\partial xy} + \frac{\partial^2 u_u}{\partial y^2} - \frac{Xu_u}{\epsilon^2 h^2}) + \frac{\mathcal{A}\eta_u}{\alpha_u}(2\frac{\partial}{\partial x}(\frac{\partial}{\partial x}(u_u - u_c)) + \frac{\partial}{\partial y}(\frac{\partial \alpha_c}{\partial x}(v_u - v_c) + \frac{\partial \alpha_u}{\partial y}(u_u - u_c))) - \frac{\mathcal{A}\eta_u\zeta\alpha_s(1-f_{sc})(u_u - u_c)}{\alpha_u h^2} - \frac{g}{n^2}\frac{u_u|u_u|}{h^{\frac{4}{3}}}] - \frac{1}{\gamma_c}C_{DG}(u_u - u_c)|\vec{u_u} - \vec{u_c}|^{J-1}$

57. $S_{y_u} = \alpha_u[g^y - \frac{\frac{1}{2}P_{b_u} h}{\alpha_f}\frac{\partial \alpha_c}{\partial y} + P_{b_u}\frac{\partial b}{\partial y} - \frac{\mathcal{A}\eta_u}{\alpha_u}(2\frac{\partial^2 u_f}{\partial y^2} + \frac{\partial^2 v_f}{\partial xy} + \frac{\partial^2 u_f}{\partial x^2} - \frac{Xu_f}{\epsilon^2 h^2}) + \frac{\mathcal{A}\eta_u}{\alpha_c}(2\frac{\partial}{\partial y}(\frac{\partial}{\partial y}(v_u - v_c)) + \frac{\partial}{\partial x}(\frac{\partial \alpha_c}{\partial y}(u_u - u_c) + \frac{\partial \alpha_c}{\partial x}(v_u - v_c))) - \frac{\mathcal{A}\eta_u\zeta\alpha_s(1-f_{sc})(v_u - v_c)}{\alpha_u h^2} - \frac{g}{n^2}\frac{v_u|u_u|}{h^{\frac{4}{3}}}] - \frac{1}{\gamma_c}C_{DG}(v_u - v_c)|\vec{u_u} - \vec{u_c}|^{J-1}$

58. $P_{b_c} = -(1-f_{sc})(1-\gamma)\frac{(1-f_{sc})\alpha_s}{(1-f_{fc})\alpha_f}g^z h - f_{sc}(1-\gamma)\frac{(f_{sc})\alpha_s}{(f_{fc})\alpha_f}g^z h$

59. $P_{b_u} = -g^z h$

60. $\gamma_c = \frac{\rho_u}{\rho_c}, \gamma = \frac{\rho_f}{\rho_s}$

61. $C_{DG} = \frac{f_{sc}\alpha_c\alpha_u(\rho_c - \rho_f)g}{U_{T,c}(\mathcal{G}(Re)) + S_p} + \frac{(1-f_{sc})\alpha_c\alpha_u(\rho_s - \rho_f)g}{U_{T,uc}(\mathcal{P}\mathcal{F}(Re_p) + (1-\mathcal{P})\mathcal{G}(Re)) + S_p}$

62. $S_p = (\frac{\mathcal{P}}{\alpha_c} + \frac{1-\mathcal{P}}{\alpha_u})\mathcal{K}$

63. $\mathcal{K} = |\alpha_c\boldsymbol{u}_c + \alpha_u\boldsymbol{u}_u|$

64. $F = \frac{\gamma}{180}(\frac{\alpha_f}{\alpha_s})^3 Re_P, \; G = \alpha_f^{M(Re_p)-1}, \; Re_p = \frac{\rho_f d U_t}{\eta_f}, \; N_R = \frac{\sqrt{gL}H\rho_f}{\alpha_f\eta_f}, \; N_{RA} = \frac{\sqrt{gL}H\rho_f}{\mathcal{A}\eta_f}$

65. $C_{Vm} = (\frac{1}{2}(\frac{1+2\alpha_c}{\alpha_u}))$

66. $\hat{\sigma} = \sigma^{\alpha\gamma}\dot{\omega}^{\beta\gamma} + \sigma^{\gamma\beta}\dot{\omega}^{\alpha\gamma} + 2G\dot{e}^{\alpha\beta} + K\dot{\epsilon}^{\gamma\gamma}\delta^{\alpha\beta} - \dot{\lambda}[9K\sin\psi\,\delta^{\alpha\beta} + \frac{G}{\sqrt{J_2}}s^{\alpha\beta}]$





67. $\dot{\lambda} = \dfrac{3\alpha K \dot{\epsilon}^{\gamma\gamma} + \left(\frac{G}{\sqrt{J_2}}\right) s^{\alpha\beta} \dot{\epsilon}^{\alpha\beta}}{27\alpha_\phi K \sin\psi + G}$
68. $K = \dfrac{E}{3(1-2v)}, G = \dfrac{E}{2(1+v)}$
69. $\sigma^{\alpha\beta} = s^{\alpha\beta} + \frac{1}{3}\sigma^{\gamma\gamma}\delta^{\alpha\beta}$
70. $\dot{\epsilon}^{\alpha\beta} = \frac{1}{2}\left(\dfrac{\partial v^\alpha}{\partial x^\beta} - \dfrac{\partial v^\beta}{\partial x^\alpha}\right) \quad \dot{\omega}^{\alpha\beta} = \frac{1}{2}\left(\dfrac{\partial v^\alpha}{\partial x^\beta} - \dfrac{\partial v^\beta}{\partial x^\alpha}\right)$
71. $\alpha_\phi = \dfrac{\tan(\phi)}{\sqrt{9+12\tan^2\phi}} \quad k_c = \dfrac{3c}{\sqrt{9+12\tan^2\phi}}$
Where X is the shape factor for vertical shearing of the fluid (X ≈ 3 in Iverson & Denlinger, 2001), $R$ is the
precipitation rate and $I$ is the infiltration rate.
**Closing the equations**
Viscosity is estimated using the empirical expression from O'Brien and Julien (1985), which relates dynamic
viscosity to the solid concentration of the fluid (Equation 72).
72. $\eta = \alpha e^{\beta\alpha_s}$
Where α is the first viscosity parameter and β the second viscosity parameter.
Finally, the settling velocity of small (d < 100 $\mu m$) grains is estimated by Stokes equations for a
homogeneous sphere in water. For larger grains ( > 1mm),the equation by Zanke (1977) is used (Equation 30).
73. $U_T = 10 \dfrac{\frac{\eta}{\rho_f}^2}{d}\left(\sqrt{1 + \dfrac{0.01\left(\frac{(\rho_s - \rho_f)}{\rho_f}gd^3\right)}{\frac{\eta}{\rho_f}}} - 1\right)$
In which $U_T$ is the settling (or terminal) velocity of a solid grain, η is the dynamic viscosity of the fluid,
$\rho_f$ is the density of the fluid, $\rho_s$ is the density of the solids, d is the grain diameter ($m$)
**1.4 Implementation in the Material Point Method numerical scheme**
Implementing the presented set of equations into a numerical scheme requires considerations of that
schemes limitations and strengths (Stomakhin et al., 2013). Fluid dynamics are almost exclusively solved using
an Eulerian finite element solution (Delestre et al., 2014; Bout et al., 2018). The diffusive advection part of such
scheme typically doesn't degrade the quality of modelling results. Solid material however is commonly
simulated with higher accuracy using an Lagrangian finite element method or discrete element method (Maurel
& Cumbescure, 2008; Stomakhin et al., 2013). Such schemes more easily allow for the material to maintain its
physical properties during movement. Additionally, advection in these schemes does not artificially diffuse the
material since the material itself is discretized, instead of the space (grid) on which the equations are solved. In
our case, the material point method (MPM) provides an appropriate tool to implement the set of presented
equations (Bui et al., 2008; Maurel & Cumbescure, 2008; Stomakhin et al., 2013). Numerous existing modelling
studies have implemented in this method (Pastor et al., 2007; Pastor et al., 2008; Abe & Kanogai, 2016). Here,
we use the MPM method to create a two-phase scheme. This allows the usage of finite elements aspects for the
fluid dynamics, which are so successfully described by the that method (particularly for water in larger areas, see
Bout et al., 2018).
**Mathematical Framework**
The mathematic framework of smooth-particles solves differential equations using discretized volumes
of mass represented by kernel functions (Libersky & Petschek, 1991; Bui et al., 2008; Stomakhin et al., 2013).
Here, we use the cubic spline kernel as used by Monaghan (2000) (Equation 74).
74. $W(r,h) = \begin{cases} \frac{10}{7\pi h^2}\left(1 - \frac{3}{2}q^2 + \frac{3}{4}q^3\right) & 0 \le |q| \ge 2 \\ \frac{10}{28\pi h^2}(2-q)^3 & 1 \le |q| < 2 \\ 0 & |q| \ge 2 \mid q < 0 \end{cases}$
Where r is the distance, h is the kernel size and $q$ is the normalized distance ($q = \frac{r}{h}$)





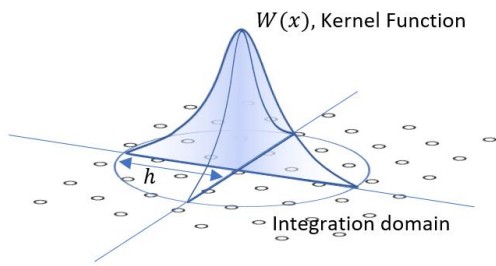


*Figure 2 Example of a kernel function used as integration domain for mathematical operations.*
Using this function mathematical operators can be defined. The average is calculated using a weighted
sum of particle values (Equation 75) while the derivative depends on the function values and the derivative of
the kernel by means of the chain rule (Equation 76) (Libersky & Petschek, 1991; Bui et al., 2008).
75. $\langle f(x) \rangle = \sum_{j=1}^{N} \frac{m_j}{\rho_j} f(x_j) W(x - x_j, h)$
76. $\langle \frac{\partial f(x)}{\partial x} \rangle = \sum_{j=1}^{N} \frac{m_j}{\rho_j} f(x_j) \frac{\partial W_{ij}}{\partial x_i}$
Where $W_{ij} = W(x_i - x_j, h)$ is the weight of particle j to particle I, $r = |x_i - x_j|$ is the distance
between two particles. The derivative of the weight function is defined by equation 77.
77. $\frac{\partial W_{ij}}{\partial x_i} = \frac{x_i - x_j}{r} \frac{\partial W_{ij}}{\partial r}$
Using these tools, the momentum equations for the particles can be defined (Equations 78-84). Here, we
follow Monaghan (1999) and Bui et al. (2008) for the definition of artificial numerical forces related to stability.
Additionally, stress-based forces are calculated on the particle level, while other momentum source terms are
solved on a Eulerian grid with spacing $h$ (identical to the kernel size).
78. $\frac{dv_i^\alpha}{dt} = \frac{1}{m_i}(F_g + F_{grid}) + \sum_{j=1}^{N} m_j \left( \frac{\sigma_i^{\alpha\beta}}{\rho_i^2} + \frac{\sigma_j^{\alpha\beta}}{\rho_j^2} + F_{ij}^n R_{ij}^{\alpha\beta} + \Pi_{ij}\delta^{\alpha\beta} \right) \frac{\partial W_{ij}}{\partial x_i^\beta}$
79. $\dot{\epsilon}^{\alpha\beta} = \frac{1}{2} \left( \sum_{j=1}^{N} \frac{m_j}{\rho_j} (v_j^\alpha - v_i^\alpha) \frac{\partial W_{ij}}{\partial x_i^\beta} + \sum_{j=1}^{N} \frac{m_j}{\rho_j} (v_j^\beta - v_i^\beta) \frac{\partial W_{ij}}{\partial x_i^\alpha} \right)$
80. $\dot{\omega}^{\alpha\beta} = \frac{1}{2} \left( \sum_{j=1}^{N} \frac{m_j}{\rho_j} (v_j^\alpha - v_i^\alpha) \frac{\partial W_{ij}}{\partial x_i^\beta} - \sum_{j=1}^{N} \frac{m_j}{\rho_j} (v_j^\beta - v_i^\beta) \frac{\partial W_{ij}}{\partial x_i^\alpha} \right)$
81. $\frac{d\sigma_{\alpha\beta}}{dt} = \sigma_i^{\alpha\gamma} \dot{\omega}_i^{\beta\gamma} + \sigma_i^{\gamma\beta} \dot{\omega}_i^{\alpha\gamma} + 2G_i \dot{e}_i^{\alpha\beta} + K_i \dot{\epsilon}^{\gamma\gamma} \delta_i^{\alpha\beta} - \dot{\lambda}_\iota \left[ 9K_i sin\psi_i \, \delta^{\alpha\beta} + \frac{G_i}{\sqrt{J_{2i}}} s_i^{\alpha\beta} \right]$
82. $\dot{\lambda}_\iota = \frac{3\alpha K \dot{\epsilon}_i^{\gamma\gamma} + \left( \frac{G_i}{\sqrt{J_{2i}}} \right) s_i^{\alpha\beta} \dot{\epsilon}_\iota^{\alpha\beta}}{27\alpha_\phi K_i sin\psi_i + G_i}$
Where $i, j$ are indices indicating the particle, $\Pi_{ij}$ is an artificial viscous force as defined by equations 83
and 84 and $F_{ij}^n R_{ij}^{\alpha\beta}$ is an artificial stress term as defined by equations 85 and 86.
83. $\Pi_{ij} = \begin{cases} \frac{\alpha_\Pi u_{sound_{ij}} \phi_{ij} + \beta_\Pi \phi^2}{\rho_{ij}} & v_{ij} \cdot x_{ij} < 0 \\ 0 & v_{ij} \cdot x_{ij} \geq 0 \end{cases}$
84. $\phi_{ij} = \frac{h_{ij} v_{ij} x_{ij}}{|x_{ij}|^2 + 0.01 h_{ij}^2}$ , $x_{ij} = x_i - x_j$ , $v_{ij} = v_i - v_j$ , $h_{ij} = \frac{1}{2}(h_i + h_j)$
85. $F_{ij}^n R_{ij}^{\alpha\beta} = \left[ \frac{W_{ij}}{W(d_0, h)} \right]^n (R_i^{\alpha\beta} + R_j^{\alpha\beta})$
86. $\overline{R_\iota^{\gamma\gamma}} = -\frac{\epsilon_0 \overline{\sigma_\iota^{\gamma\gamma}}}{\rho_i^2}$
Where $\epsilon_0$ is a small parameter ranging from 0 to 1, $\alpha_\Pi$ and $\beta_\Pi$ are constants in the artificial viscous
force (often chosen close to 1), $u_{sound}$ is the speed of sound in the material.



The conversion from particles to gridded values and reversed depends on a grid basis function that
weighs the influence of particle values for a grid center. Here, a function derived from dyadic products of one-
dimensional cubic B-splines is used as was done by Steffen et al. (2008) and Stomakhin et al. (2013) (Equation
84).

87. $N(x) = N(x^x) * N(x^y), \quad N(x) = \begin{cases} \frac{1}{2}|x|^3 - x^2 + \frac{2}{3} & 0 \le |x| \ge 2 \\ -\frac{1}{6}|x|^3 + x^2 - 2|x| + \frac{4}{3} & 1 \le |x| < 2 \\ 0 & |x| \ge 2 \mid x = 0 \end{cases}$

### Particle placement

Particle placement is typically done in a constant pattern, as initial conditions have some constant
density. The simplest approach is a regular square or triangular network, with particles on the corners of the
network. Here, we use an approach that is more adaptable to spatially-varying initial flow height. The $R_2$
sequence approaches, with a regular quasirandom sequence, a set of evenly distributed points within a square
(Roberts, 2020) (Equation 85).
88. $x_n = n\boldsymbol{\alpha} \bmod 1 , \quad \boldsymbol{\alpha} = \left(\frac{1}{c_p}, \frac{1}{c_p^2}\right)$
Where $x_n$ is the relative location of the $n^{\text{th}}$ particle within a gridcell, $c_p = \left(\frac{9+\sqrt{69}}{18}\right)^{\frac{1}{3}} + \left(\frac{9-\sqrt{69}}{18}\right)^{\frac{1}{3}} \approx$
1.32471795572 is the plastic constant.

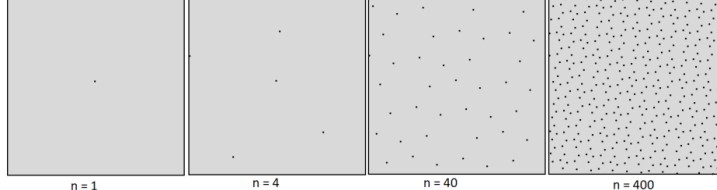


*Figure 3 Example particle distributions using the $R_2$ sequence, note that, while not all particles are*
*equidistant, the method produces distributed particle patterns that adapt well to varying density.*
The number of particles placed for a particular flow height depends on the particle volume $V_I$, which is
taken as a global constant during the simulation.

## 2   Flume Experiments
### 2.1 Flume Setup

In order to validate the presented model, several controlled experiments were performed and reproduced
using the developed equations. The flume setup consists of a steep incline, followed by a near-flat runout plane
(Figure 3). On the separation point of the two planes, a massive and attached obstacle is present that blocks the
path of two fifth of the moving material. For the exact dimensions of both the flume parts and the obstacle, see
figure 3.

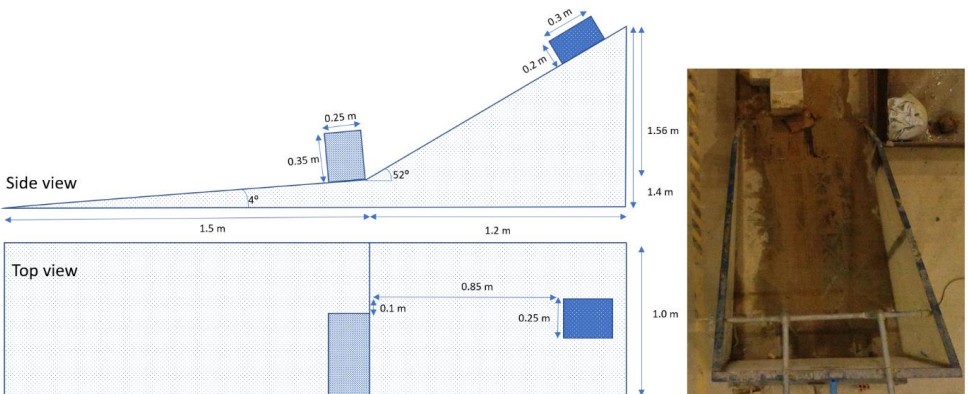

*Figure 4 The dimensions of the flume experiment setup used in this work.*
Two tests were performed whereby a cohesive granular matrix was released at the upper part of the
flume setup. Both of these volumes had dimensions of 0.2x0.3x0.25 meter (height,length,width). For both of
these materials, a mixture high-organic content silty-clay soils where used. The materials strength parameters
were obtained using tri-axial testing (Cohesion, internal friction angle Youngs modulus and Poisson Ration. The
first set of materials properties where $c = 26.7$ kPa and $\phi = 28°$. The second set materials properties where $c = $
18.3 kPa and $\phi = 27°$. For both of the events, pre-and post release elevations models were made using
photogrammetry. The model was set up to replicate the situations using the measured input parameters.
Numerical settings were chosen as $\{\alpha_s = 0.5, \alpha_f = 0.5, f_{sc} = 1.0, f_{fc} = 1.0, \rho_f = 1000, \rho_s = 2400, E = 12 \cdot$
$10^6 Pa, K = 23 \cdot 10^6 Pa, \psi = 0, \alpha_\Pi = 1, \beta_\Pi = 1, X, \zeta, j = 2, u_{sound} = 600, dx = 10, V_l = , h = 10, n = $
$0.1, \alpha = 1, \beta = 10, M = 2.4, \mathcal{B} = 0, N_R = 15000, N_{RA} = 30\}$. Calibration was performed by means of input
variation. The solid fraction, and elastic and bulk modulus were varied between 20 and 200 percent of their
original values with increments of 10 percent. Accuracy was assessed based on the percentage accuracy of the
deposition (comparison of modelled vs observed presence of material).
**2.2 Results**
Both the mapped extent of the material after flume experiments, as the simulation results are shown in
figure 5. Calibrated values for the simulations are $\{\alpha_s = 0.45, E = 21.6 \cdot 10^6 Pa, K = 13.8 \cdot 10^6 Pa\}$.



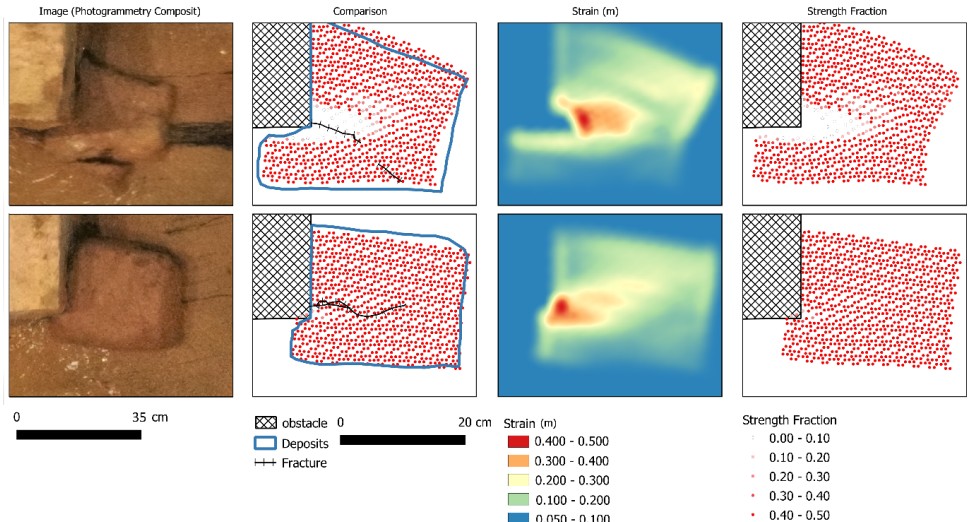


*Figure 5 A comparison of the final deposits of the simulations and the mapped final deposits and cracks within the material. From left to right: Photogrammetry mosaic, comparison of simulation results to mapped flume experiment, strain, final strength fraction remaining.*

As soon as the block of material impacts the obstacle, stress increases as the moving objects is deformed. This stress quickly propagates through the object. Within the scenario with lower cohesive strength, as soon as the stress reached beyond the yield strength, degradation of strength parameters took place. In the results, a fracture line developed along the corner of the obstacle into the length direction of the moving mass. Eventually, this fracture developed to half the length of the moving body and severe deformation resulted. As was observed from the tests, the first material experienced a critical fracture while the second test resulted in moderate deformation near the impact location. Generally, the results compare well with the observed patters, although the exact shape of the fracture is not replicated. Several reasons might be the cause of the moderately accurate fracture patterns. Other studies used a more controlled setup where uncertainties in applied stress and material properties where reduced. Furthermore, the homogeneity of the material used in the tests can not completely assumed. Realistically, minor alterations in compression used to create the clay blocks has left spatial variation in density, cohesion and other strength parameters.

## 3. Numerical Tests

### 3.1 Numerical Setup

In order to further investigate some of the behaviors of the model, and highlight the novel types of mass movement dynamics that the model implements, several numerical tests have been performed. The setup of these tests is shown in figure 6.






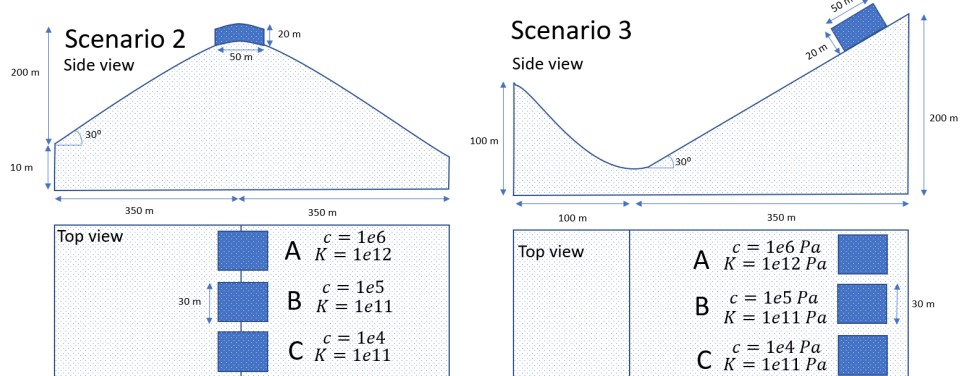

*Figure 6 The dimensions of the numerical experiment setups used in this work. Setup 1 (left) and Setup 2 (right)*

Numerical settings were chosen for three different blocks with equal volume but distinct properties.
Cohesive strength and the bulk modulus were varied (see figure 6). Remaining parameters were chosen as
$\{\alpha_s = 0.5, \alpha_f = 0.5, f_{sc} = 1.0, f_{fc} = 1.0, \rho_f = 1000 \, kgm^{-3}, \rho_s = 2400 \, kgm^{-3}, E = 1e12 \, Pa, \psi = 0, \alpha_\Pi =$
$1, \beta_\Pi = 1, X, \zeta, j = 2, u_{sound} = 600 \, ms^{-1}, dx = 10 \, m, V_I, h = 10 \, m, n = 0.1, \alpha = 1, \beta = 10, M = 2.4, \mathcal{B} =$
$0, N_R = 15000, N_{RA} = 30\}$.
**3.1 Results**

Several time-slices for the described numerical scenarios are shown in figure 7 and 8.


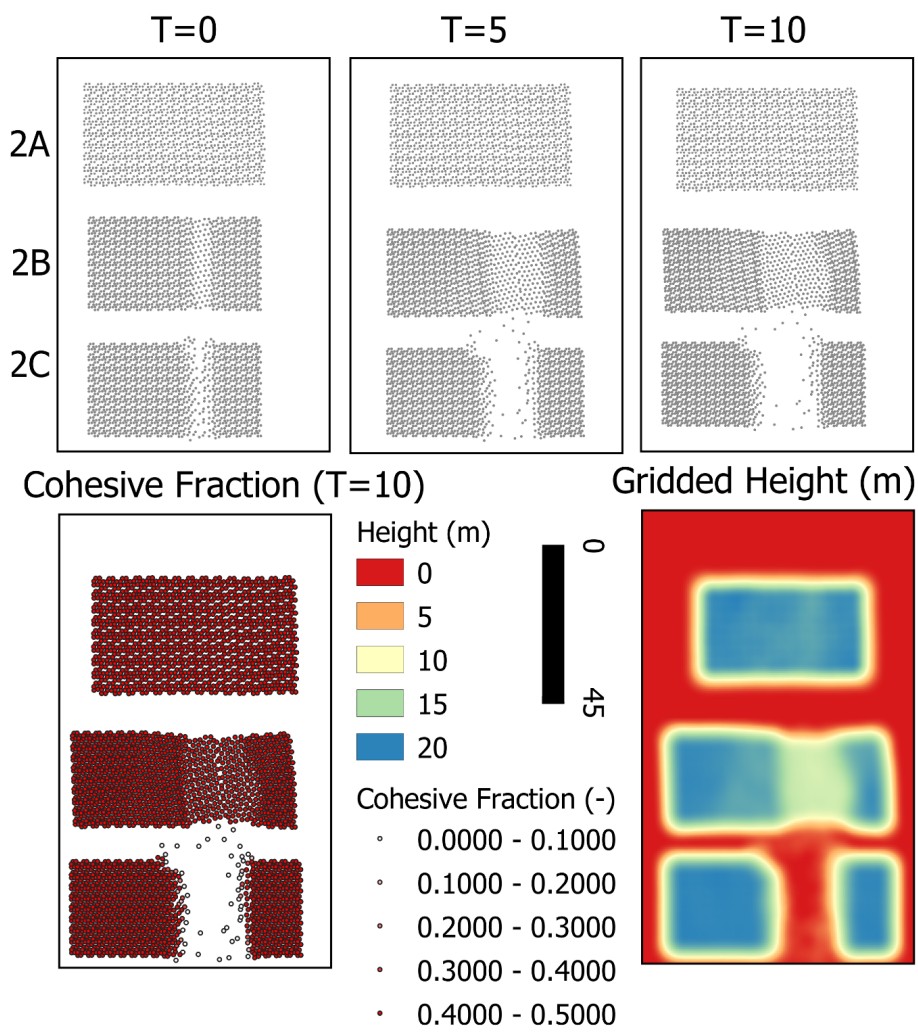


Figure 7 Several time-slices for numerical scenarios 2(A/B/C). See figure 6 for the dimensions and
terrain setup.

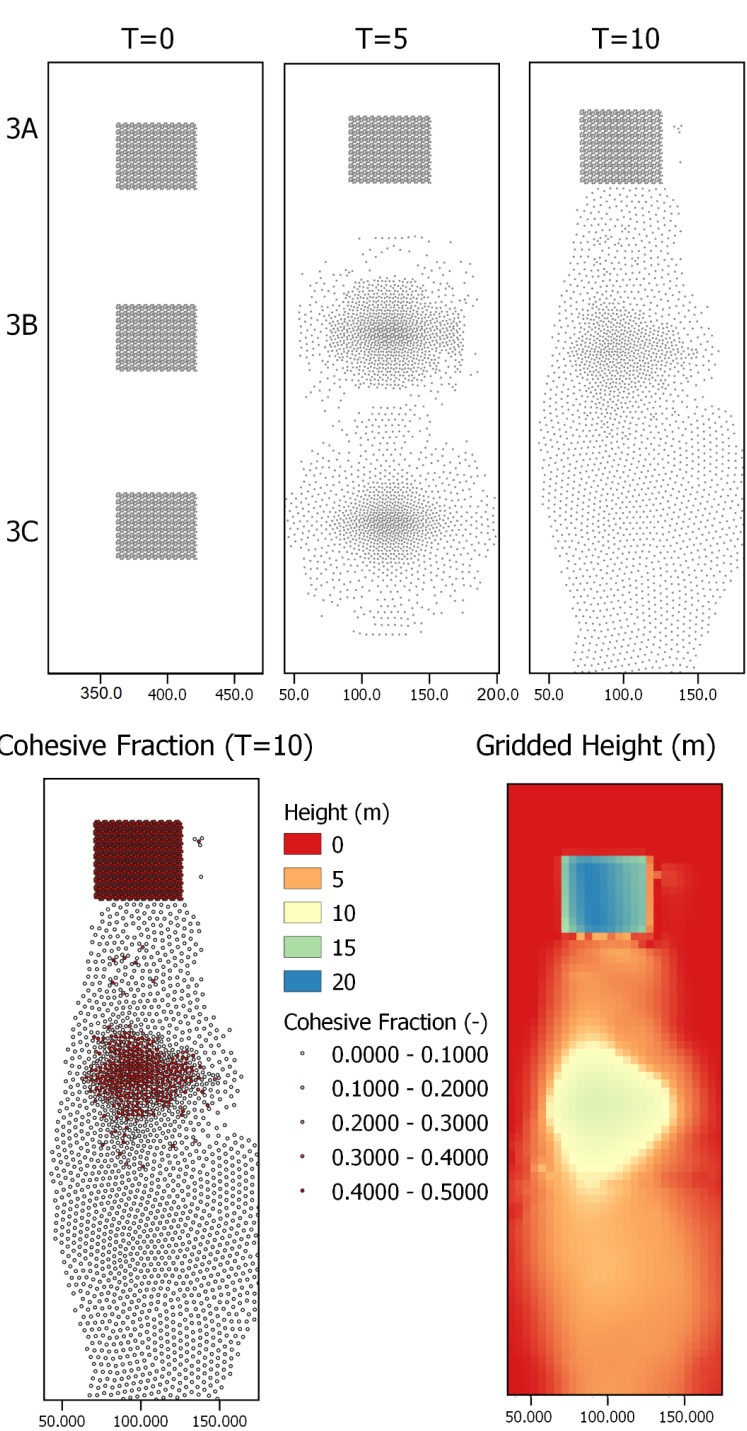

*Figure 8 Several time-slices for numerical scenarios 3(A/B/C). See figure 6 for the dimensions and terrain setup.*

Fractures develop in the mass movements based on acceleration differences and cohesive strength. For
scenario 2A, the stress state does not reach beyond the yield surface, and all material is moved as a single block.
Scenario 2B, which features lowered cohesive strength, fractures and the masses separate based on the
acceleration caused by slopes.

Fracturing behavior can occur in MPM schemes due to numerical limitations inherent in the usage of a
limited integration domain. Here, validation of real physically-based fracturing is present in the remaining
cohesive fraction. This value only reduces in case of plastic yield, where increasing strain degrades strength
parameters according to our proposed criteria. Numerical fractures would thus have a cohesive fraction of 1. In
all simulated scenarios, such numerical issues were not observed.

Fragmentation occurs due to spatial variation in acceleration in the case of scenario 3A and 3B. For
scenario 3A, the yield surface is not reached and the original structure of the mass is maintained during
movement. For 3C, fragmentation is induced be lateral pressure and buoyancy forces alone. Scenario 3B
experiences slight fragmentation at the edges of the mass, but predominantly fragments when reaching the
valley, after which part of the material is accelerated to count to the velocity of the mass. For all the shown
simulations, fragmentation does not lead to significant phase separation since virtual mass and drag forces
converge the separate phase velocities to their mixture-averaged velocity. The strength of these forces partly
depends on the parameters, effects of more immediate phase-separation could by studied if other parameters are
used as input.

**4. Discussion**

A variety of existing landslide models simulate the behavior of lateral connected material through a
non-linear, non-Newtonian viscous relationship (Boetticher et al., 2016; Fornes et al., 2017; Pudasaini &
Mergili, 2019). These relationships include a yield stress and are usually regularized to prevent singularities from
occurring. While this approach is incredibly powerful, it is fundamentally different from the work proposed here.
These viscous approaches do not distinguish between elastic or plastic deformation, and typically ignore
deformations if stress is insufficient. Additionally, fracturing is not implemented in these models. The approach
taken in this work attempts to simulate a full stress-strain relationship with Mohr-Coulomb type yield surface.
This does provides new types of behavior and can be combined with non-Newtonian viscous approaches as
mentioned above. A major downside to the presented work is the steep increase in computational time required
to maintain an accurate and stable simulation. Commonly, an increase of near a 100 times has been observed
during the development of the presented model.

The presented model shows a good likeness to flume experiments and numerical tests highlight
behavior that is commonly observed for landslide movements. There are however, inherent scaling issues and the
material used in the flume experiments is unlikely to form larger landslide masses. The measured physical
strength parameters of the material used in the flume experiments would not allow for sustained structured
movement at larger scales. There is thus the need for more, real-scale, validation cases. The application of the
presented type of model is most directly noticeable for block-type landslide movements that have fragmented
either upon impact of some obstacle or during transition phase. Of importance here is that the moment of
fragmentation is often not reported in studies on fast-moving landslides, potentially due to the complexities in
knowing the details on this behavior from post-event evidence. Validation would therefore have to occur on
cases where deposits are not fully fragmented, indicating that this process was ongoing during the whole
movement duration. The spatial extent of initiation and deposition would then allow validation of the model.
Another major opportunity for validation of the novel aspects of the model is the full three-dimensional
application to landslides that were reported to have lubrication effects due to fragmentation of lower fraction of
flow due to shear.

An important point of consideration in the development of complex multi-process generalized models is
the applicability. As a detailed investigative research tool, these models provide a basic scenario of usage.
However, both for research and beyond this, in applicability in disaster risk reduction decision support, the
benefit drawn from these models depends on the practical requirement for parameterization and the
computational demands for simulation. With an increasing complexity in the description of multi-process
mechanics comes the requirement of more measured or estimated physical parameters. Inspection of the
presented method shows that in principle, a minor amount of new parameters are introduced. The cohesive
strength, a major focus of the model, becomes highly important depending on the type of movement being
investigated. Additionally, the bulk and elastic modulus are required. These three parameters are common
simulation parameters in geotechnical research and can be obtained from common tests on sampled material
(Alsalman et al., 2015). Finally, the basal pressure propagation parameter ($\mathcal{B}$) is introduced. However, within
this work, the value of this parameter is chosen to have a constant value of one. As a results, the model does
require additional parameters, although these are relatively easy to obtain with accuracy.





There are a variety of aspects of the model that could be significantly improved. Here, we list several
major opportunities of future research.

**1) Groundwater mechanics**

The presented model allows for the a solid or granular matrix to be present within the flow. We have assumed the flows in and out of these matrices are sufficiently small to be ignored. In reality, there is a fluid flux in and out of structured solids. This could occur both due to pressure differences as due to stress and strain of the structured solids. Implementing this kind of mechanics requires a dynamic, solid-properties dependent, soil water retention curve (Van Looy et al., 2017). An example of MPM soil mechanics with dynamic groundwater implementation can be found in Bandera et al. (2016).

**2) Implementing Entrainment and Deposition**

Current equations for entrainment (erosion with major grain-grain interactions) is limited to unstructured mixture flows (Iverson, 2012; Iverson & Ouyang, 2015; Pudasaini & Fischer, 2016). Extending these models to include a contribution from structured solids would be required to implement entrainment in the presented work.

**3) Separation of phases**

A major assumption in the presented work is that the velocities of structured solids, free solids and confined fluids are all equal. In reality, there might be separation of structured and free solids phases. Additionally, we already discussed the possibility of in-and outflux of confined fluids from the solid matrix. Recent innovations on three-phase mixture flows might be used to extend the presented work to a three, four or five-phase model by separating free solids, confined fluids or adding a Bingham-viscous solid-fluid phase (Pudasaini & Mergili, 2019). However, while this would implement an additional process, it would significantly increase complexity of the equations (in an exponential manner with relation to the number of phases) and the numerical solutions which could hinder practical applicability.

**4) Application to large, slow moving landslides.**

When confined fluids would act as a distinct phase, guided by the mechanics of water flow in granular matrix, ground water pressures and movement through the structured solids could be described. This might enable the model to do detailed deformation/groundwater simulation of large slow-moving landslides.

**5) Numerical Improvements**

Numerical techniques for particle-based discretized methods (SPH, MPM) have been proposed in the literature. A common issue is numerical fracturing of materials when particle strain increases beyond the length of the kernel function. Then, the connection between particles is lost and fracturing occurs as an artifact of the numerical method. This issue is partly solved by the artificial stress term as is also used by Bui et al. (2008). Additionally, geometric subdivide, as used by Xu et al. (2012) and Li et al. (2015), could counter these artificial fractures. Implementing this technique does require additional work to maintain mass and momentum conservation.

**6) Three-dimensional solutions**

In a variety of scenarios, the assumptions made in depth-averaged application of flow models are invalid. A common example is the impact of mass movements into lakes, or other large water bodies. In such cases, the vertical velocity and concentration variables are not well-described by their depth-averaged counterparts. Additionally, the lubrication effect of basal fragmentation of landslides due to shear can not be described without velocity-profiles and a vertical stress-solution. Full three-dimensional application would therefore have the potential to increase understanding on these important processes.

**5. Conclusions**

We have presented a novel generalized mass movement model that can describe both unstructured mixture flows and Structured movements of Mohr-Coulomb type material. The presented equations are part of the continuous development of the OpenLISEM Hazard model, an open-source tool for physically-based multi-hazard simulations. The model builds on the works of Pudasaini (2012) and Bui et al. (2008) to develop a single holistic set of equations. The model was implemented in a GPU-based Material Point Method (MPM) Code. The equations were validated on flume experiments and numerical tests, that highlight the new movement dynamics possible with the presented model. The integration of cohesive structure and a full stress-strain relationship for the structured solids allows for movement of block-type slides as a single whole. Interactions with terrain, other flow masses or obstacles lead to elastic-plastic deformation and eventually fragmentation. This type of self-alteration of flow properties is novel with mass movement models. Although the presented equations can provide additional detail for specific mass movement types, applicability of the model for real events need to be investigated as computational costs are significantly increased.



The presented simulation both validate the basic behavior of the model, as well as highlight the types of
flow dynamics made possible by the presented equations. The models dependency of breaking to cohesive
strength and internal friction angle matches the flume experiments. The numerical examples show commonly-
described behavior for landslide movements. Although the simulations compare well to the flume experiments,
validation is required for real-scale application to various types of mass movements. Additionally, the presented
equations still lack descriptions of processes that might become important. Separating the fluid and solid phases
such as done by Pudasaini & Mergili (2019), could improve flow dynamics and phase separation. With added
ground-water mechanics, such as done in Bandera et al. (2016), slow-moving landslide simulations might be
described.
**6. Code and Data Availability**

All code and data used within this work are made open-source as part of the continuous development of
the OpenLISEM Hazard model under the GNU General Public Licence v3.0. The code and the data are hosted
on Github (https://github.com/bastianvandenbout/OpenLISEM-Hazard-2.0-Pre-Release). Both binaries
and a copy of the source code are also available on Sourceforge, where the manual and compilation guide can
similarly be found (https://sourceforge.net/projects/lisem/). Finally, more information can be found at the blog
(https://blog.utwente.nl/lisem/)

The software, and its user interface, are written for windows, but platform independent libraries are
used and compilation might be performed on other platforms.
Hardware requirements for the usage of the model are a 64-bit Operating system that can compile all required
external libraries (see the manual for a full list and description). A graphical processing unit conforming to at
least the OpenCL 1.2 standard and support for both OpenGL 4.2 and OpenGL/OpenCL interoperability.
Additionally, an approximate 500 mb of hard drive space and 750 mb of memory must be available.



### Appendix A. List of Symbols


$h$ is the flow height
$s$ is the solid phase
$f$ is the fluid phase
$sc$ is the structured solid phase
$fc$ is the confined fluid phase
$\rho_f$ is the density of fluids
$\rho_s$ is the density of solids
$\alpha_f$ is the volumetric fluid phase fraction
$\alpha_s$ is the volumetric solid phase fraction
$f_{sc}$ is the fraction of solids that is structured (confining)
$f_{fc}$ is the fraction of fluids that is confined
$\alpha_c$ is the volumetric fraction of solids, structured solids and confined fluids
$\alpha_u$ is the volumetric fraction of free fluids (unconfined phase).
$\rho_{sc}$ is the volume-averaged density of the solids and confined fluids
$\boldsymbol{u_u}$ is the velocity of the unconfined phase (free fluids)
$\boldsymbol{u_c}$ is the velocity of the solids, confining solids and confined fluids
$\boldsymbol{u_s}$ is the velocity of the solids
$\boldsymbol{f}$ is the body force
$\boldsymbol{M}_{DG}$ is the drag force
$\boldsymbol{M}_{vm}$ is the virtual mass force
$\boldsymbol{T}_c$ is the stress tensor for eh solids, confining solids and confined fluids
$\boldsymbol{T}_u$ is the stress tensor for the free fluid phase
$\boldsymbol{\sigma}$ is the stress tensor
$\dot{s}$ is the deviatoric shear stress rate tensor
$\delta$ is the Kronecker delta
$\dot{\epsilon}_{plastic}$ is the plastic strain rate
$\dot{\epsilon}_{elastic}$ is the elastic strain rate
$\lambda$ is the plastic multiplier rate
$g$ is the plastic potential function
$\dot{\epsilon}_{total}$ is the total strain rate
$\dot{e}$ is the deviatoric strain rate
$\nu$ is Poisson's ratio
$E$ is the elastic Young's Modulus
$G$ is the shear modulus
$K$ is the Bulk elastic modulus
$f(I_1, J_2)$ is the yield surface, or yield criterion
$g(I_1, J_2)$ is the plastic potential function
$\psi$ is the dilatancy angle
$I_1$ is the first stress invariant
$J_2$ is the second stress invariant
$\alpha_\phi$ is the first Ducker-Prager material constant
$k_c$ is the second Ducker-Prager material constant
$\dot{\omega}$ is the spin rate tensor
$\epsilon_{v0}$ is the initial volumetric strain
$\epsilon_v$ is the volumetric strain
$c_0$ is the initial cohesion
$\boldsymbol{\tau}_f$ is the fluid Gauchy stress tensor
$P_f$ is the fluid pressure
$\eta_f$ is the fluids dynamic viscosity
$\mathcal{A}$ is the mobility of the fluid at the interface
$\mathcal{C}_{DG}$ is the drag coefficient
$U_{T,c}$ is the settling velocity of the solids, structured solids and confined fluids
$U_{T,uc}$ is the settling velocity of the unstructured solids
$\mathcal{F}$ is the drag contribution from solid-like drag
$\mathcal{G}$ is the drag contribution from fluid-like drag
$S_p$ is the smoothing function
$\mathcal{K}$ is the absolute total mass flux





$M(Re_p)$ is an empirical function weakly dependent on the Reynolds number
$\mathcal{P}$ the partitioning parameter for the fluid and solid like contributions to drag
$m$ is an exponent for $\mathcal{P}$
$C_{VMG}$ is the virtual mass coefficient
$|\boldsymbol{S}|$ is the norm of the shear force
$N$ is the normal force on a plane element
$g$ is the gravitational acceleration
$P_{b_{s,u}}$ is the basal pressure from
$P_{b_u}$ is the basal pressure from the free fluids
$P_{b_c}$ is the basal pressure from the solids, structured solids and confined fluids
$\mathcal{B}$ is the pressure propagation factor for structured solids
$K_a$ is the active lateral earth pressure coefficient
$K_p$ is the passive lateral earth pressure coefficient
$\zeta$ is a shape factor for the vertical gradient in solid concentration
$n$ is Mannings surface roughness coefficient
X is the shape factor for the vertical fluid velocity profile
$Re_p$ is the particle Reynolds Number
$N_R$ is the Reynolds Number
$N_{RA}$ is the interfacial Reynolds Number
$H$ is the typical height of the flow
$L$ is the typical length of the flow
α is the first viscosity parameter
β the second viscosity parameter
d is the grain diameter
$W$ is the kernel weight function
$r$ is the distance
$h$ is the kernel width (not to be confused with the flow height)
$q$ is the normalized particle distance
$\Pi_{ij}$ is an artificial viscosity term
$F_{ij}^n R_{ij}^{\alpha\beta}$ is an artificial stress term
$\epsilon_0$ is a constant parameter for the artificial stress term
$\alpha_\Pi$ and $\beta_\Pi$ are constants in the artificial viscous force
$u_{sound}$ is the speed of sound in the material
$N(\boldsymbol{x})$ is the Grid-kernel function
$c_p$ is the plastic coefficient







**Appendix B. Stress Remapping**
If, either due to degradation of strength parameters, or building numerical errors, the state of the stress
tensor lies beyond the yield surface, a correction must be applied. We implement the correction scheme used by
Bui et al. (2008). This scheme considers two primary ways in which the stress can have an undesired state:
Tension cracking, and imperfectly plastic stress.
**Tension Cracking**
In the case of tension cracking, the stress state has moved beyond the apex of the yield surface, as
described by Chen & Mizuno (1990). The employed solution in this case is to re-map the stress tensor along the
$I_1$ axis to be at this apex. The apex is provided by the yield function (Equation 89)
89.  $-\alpha_\phi I_1 + k_c < 0$
To solve for this condition, the non-deviatoric stress state is increased (since $I_1 - \frac{k_c}{\alpha_\phi}$ is negative) to lie
perpendicular to the apex point on the $I_1$ axis (Equation ).
90.  $\widetilde{\sigma^{\gamma\gamma}} = rs^{\gamma\gamma} - \frac{1}{3}\left(I_1 - \frac{k_c}{\alpha_\phi}\right)$
**Imperfect Plastic Stress**
Imperfect plastic stress described the state where the stress tensor lies above the apex, but beyond the
yield criterion, thus have more stress than supported by the failure criteria that is set. This criteria is simply the
yield surface itself (Equation 91).
91.  $-\alpha_\phi I_1 + k_c < \sqrt{J_2}$
For this state, re-mapping is done by scaling of the $J_2$ value (Equations 92, 93 and 94).
92.  $r = \frac{-\alpha_\phi I_1 + k_c}{\sqrt{J_2}}$
93.  $\widetilde{\sigma^{\gamma\gamma}} = rs^{\gamma\gamma} + \frac{1}{3}I_1$
94.  $\widetilde{\sigma^{xy}} = rs^{xy}, \widetilde{\sigma^{xy}} = rs^{xz}, \widetilde{\sigma^{xy}} = rs^{yz}$



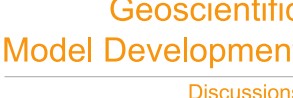
**Appendix C. Software Implementation**
The model presented in this article is part of the continued development of the OpenLISEM modelling
tools. The most recent set of equations of implemented in the open-source alpha version of OpenLISEM Hazard
2. Here, we describe the details of the implementation of the model into software.
**Hybrid MPM**
We utilize the MPM framework to be able to discretize part of the equations on a Eulerian regural grid,
and part of the equations on the Lagrangian particles. Our distinct take on this method is the representation of the
fluid phase completely as a finite element solution, while solids are simulated as discrete particle volumes. This
allows the model to use the major benefits that are present when depth-averaged fluid flow is simulated in a grid.
Both numerical efficiency, and high-accuracy coupling with hydrology are lacking in particle methods. For the
solid phase, non-dissapative advection, fracturing and stiffness is a major benefit of the MPM approach. Since
our model assumed confined fluids share their velocity with the solids, we advect the confined fluids as part of
the particles. Total fluid volume is then calculated from the free fluids in the finite element data, and the gridded
particle data. A flowchart of the software setup is provided in figure 6.

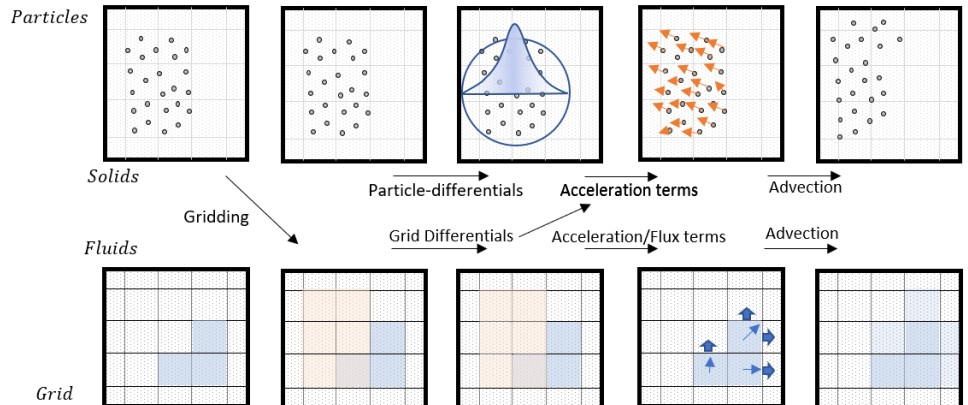

*Figure 9 The sub-steps taken by the software to complete a single step of numerical integration.*
**Finite element solution**
We use a regular cartesian grid to describe the modelling domain. Terrain and cell-boundary based
variables are re-produces using the MUSCL piecewise linear reconstruction (Delestre et al., 2014). For each cell-
boundary, a left and right estimation of acceleration terms, velocity updates and new discharges is made. The left
estimates use left-reconstructed variables while the other uses right-reconstructed variables. The final average
flux through the boundary determines actual mass and momentum transfer. Local acceleration is averaged from
the right estimate of the left boundary and left estimate of the right boundary. An additional benefit of the used
scheme is the automatic estimation of continuous and discontinuous terrain. The piecewise linear reconstructions
do not guarantee smooth terrain, for sharp locally variable terrain, pressure terms from vertical walls arise that
block momentum. These terms allow for better estimation of momentum loss by barriers, but can be turned off if
required for the simulated scenario.

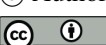


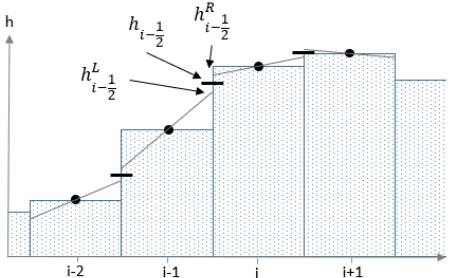


856 *Figure 10 Piecewise linear reconstruction is used by the MUSCL scheme to estimate values of flow*
857 *heights, velocities and terrain at cell-boundaries.*

858 **GPU acceleration using OpenCL/OpenGL**

859 In order to create a more efficient setup, both the finite element and particle interactions are performed
860 on the GPU. We utilize the OpenCL API to compile kernels written in c-style language. These kernels are
861 compiled at the start of the simulation, and thereby allow for easy customization by users. While the usage of
862 OpenCL 1.1 forces the usage of single precision floating point numbers, it allows for a wider range of GPU types
863 to be supported. Finite element solutions on the GPU are straightforward, as maps are a basic data storage type
864 for graphical processing units. Particles are stored as single-precision floating point arrays. Within the
865 framework of MPM, iteration of particles within a kernel is required for each timestep and particle. This
866 effectively means $O(n^2)$ operations are required. Significant efficiency improvements are obtained by pre-
867 calculation sorting. Particles are sorted based on their location within the finite element grid. Based on the id of
868 the gridcell, a bitonic mergesort is performed. This sorting algorithm works seamlessly on parallel architecture
869 and operates as $O(nlog^2(n))$ (Batcher, 1968). The then, a raster is allocated to store the first indexed occurrence
870 within the sorted list of particles of that gridcell. Since the kernel used for the presented work extends at most to
871 a full width of two gridcells, we must iterate over all particles present in 9 neighboring grid cells.

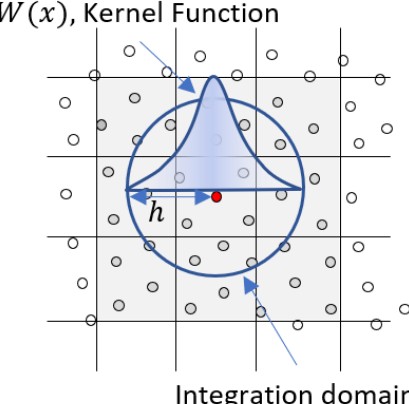

873 *Figure 11 By limiting the kernel with and sorting particles before calculation, only the distance of*
874 *particles in neighboring cells need to be checked, significantly reducing computational load, particularly for*
875 *larger datasets.*

876 A final benefit to the usage of OpenCL is direct access to simulation variables for visualization in
877 OpenGL using the OpenGL/OpenCL interoperability functionality. The built-in viewing window of OpenLISEM
878 Hazard 2.0 alpha directly uses the data to draw both particles, shapefiles and grid data using customizable
879 shaders written in the openGL shader language.



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
