# Peer review of "OpenLISEM Hazard model 2.0a"

_Geoscientific Model Development, 2020_

## Referee Comment (RC1) · Anonymous Referee #1 · 4 Jul 2020

The proposed paper, even if interesting, is not acceptable in the present form. Main deficiencies are:

1) The introduction is not clear about the scope of the work and its application. The writer means the unstructured and structured flows. The writer suggests to initially describe the phenomena, highlighting the structured and unstructured flows. After that, models usually implemented to simulate such flows can be discussed. At the end of this discussion, the new modeling approach should be introduced. In the last part of the introduction, authors claim the introduction of a model simulating structured flow followed by simulation of the fragmentation, that is an unstructured flow: this aspect is not clear. 2) In Section 1 it is described the dynamics of landslide or rock failure extending this behavior to all the gravitational flows. Indeed, debris flows, even named in

the section title "set of debris flow equations incorporating internal structure" and being a gravitational flow, are not part of the phenomena described at lines 78-101. Really, most of debris flows are runoff-generated debris flows: large quantities of sediments are entrained into runoff (Coe et al., 2008; Kean et al., 2013; Hurlimann et al., 2014):. In other words, the shear stress exerted by stream flow causes the entrainment of the sediment. Conversely, landslide-induced debris flows (Iverson et al., 1997; Iverson, 1997) are very few. 3) Figure 1: the colors difference between fragmented, free and confined fluid is not clear. 4) After reading all the paper, the writer summarizes that it concerns the behaviour of blocks in a cohesive matrix. Therefore, the model here presented cannot cover all the mass movements. The authors should explicitly write that it concerns only a class of mass movements. The writer has some concern about its applicability to granular flow. A model that simulates the flow of a cohesive matrix around an obstacle was also presented by Greco et al. (2019).

The writer suggests the partial rewriting of the manuscript according to the issues raised above

The following are the detailed comments and errors.

Line 12: main difference between debris-flow and landslide is the fluid content that rules the rheology of the flow

Line 14 "However, models commonly assume unstructured and fragmented flow after initiation of movement." Unclear sentence for a reader without knowledge on structured and unstructured flow: rewrite it as "Such type of models assumes an unstructured flow: explanation . ...."

Line 24 "ground-water flow descriptions"???

Lines 55-58 "However, this approach lacks the process of fragmentation and internal failure. Thus, within current mass movement models, there might be improvements available from assuming non-fragmented movement. This would allow for description

of structured mass movement dynamics." Unclear period. Moreover, it is not well linked to previous period at lines 49-54.

Lines 507-508 "On the separation point of the two planes, a massive and attached obstacle is present that blocks the path of two fifth of the moving material. " the expression "a massive and attached obstacle " is not well suited. Perhaps it is better "A massive obstacle is placed on the separation point of the two planes . . . . . ." Moreover, which is the sense of "that blocks the path of two fifth of the moving material". Perhaps this obstacle blocks two fifth of the flow width. Finally, in figure 4 it seems blocking three fifths rather than two fifths.

Line 632. There are also the approaches of Medina et al., 2008; Armanini et al., 2009, Frank et al., 2015, Cuomo et al., 2016 and Gregoretti et al., 2019.

References Armanini, A., Fraccarollo, L., Rosatti, G., 2009. Two-dimensional simulation of debris flows in erodible channels. Comput. Geosci. 35 (5), 993–1006.

Coe, J.A., Kinner, D.A., Godt, J.W., 2008. Initiation conditions for debris flows generated by runoff at Chalk Cliffs, central Colorado. Geomorphology 96, 270–297.

Cuomo, S., Pastor, M., Capobianco, V., Cascini, L., 2016. Modelling the space- time bed entrainment for flow-like landslide. Eng. Geol. 212, 10–20.https://doi.org/10.10116/j.enggeo.2016.07.011

Frank, F., McArdell, B.W., Huggel, C., Vieli, A., 2015. The importance of entrainment and bulking on debris flow runout modeling: examples from the Swiss. Alps. Nat. Hazards Earth Syst. Sci. 15, 2569–2583. https://doi.org/10.5194/nhess-15-2569-2015

Greco, M., Di Cristo, C., Iervolino, M., 2019. Numerical simulation of mud-flows impact-ing structures. J. Mt. Sci. 16 (2). https://doi.org/10.1007/s11629-018-5279-5.

Hurlimann, M., Abanco, C., Moya, J., Vilajosana, I., 2014. Results and experiences gathered at the Rebaixader debris-flow monitoring site, Central Pyrenees, Spain. Landslides 161–175. https://doi.org/10.1007/s10346-013-0452-y.

Iverson (1997) The physics of debris flows Reviews of Geophysics, 35, 3 /August 1997 pages 245–296

Iverson RM, Reid ME, Lahusen RG. 1997. Debris-flow mobilization from landslides. Annual Review of Earth and Planetary Sciences 25: 85–136.

Kean, J.W., McCoy, S.W., Tucker, G.E., Staley, D.M., Coe, J.A., 2013. Runoff-generated debris flows: observations and modeling of surge initiation, magnitude and frequency. J. Geophys. Res. 118, 2190–2207. https://doi.org/10.1029/jgrf20148.

Medina, V., Hurlimann, M., Bateman, A., 2008. Application of FLATModel, a 2D a finite volume code to debris flows in the northeastern part of the Iberian Peninsula.

---

## Referee Comment (RC2) · Anonymous Referee #2 · 14 Sep 2020

The authors present a model to describe mass movements of gravity-driven flows such as debris flows and landslides. Understanding the dynamics of these types of flow is important to mitigate the hazards associated with them as well as the details of river channel dynamics to which they supply a substantial amount of sediment. Much of the community understanding of these flows are drawn from experimental and in-field studies and limited numerical models. Therefore, developing a numerical model to precisely model these types of flows can enhance our understanding regarding their dynamics. However, at this stage, these numerical models can be computationally expensive as authors have mentioned in the discussion of this manuscript, and may not be practical. But this should not be a limit in developing numerical models to describe these types of flow. With that being said, I believe this manuscript and the model

developed by authors is deserved to get published after addressing a few things as mentioned below. Therefore, I recommend a minor revision.

1) The introduction is not labeled. I suggest labeling the "introduction" section as 1, then describe the model in section 2 and so on. This would then be consistent with the outline of the paper in lines 69 to 75. 2) The title in line 76 ("A set of debris flow equations....) should be changed. The description given in lines 78 to 101 is better related to other types of hillslopes transport such as rock avalanches and landslides rather than debris flows. 3) Line 49: remove "the" 4) Line 69-70: "an arbitrary" 5) Line 107: the volume fraction for solid and fluid are not defined correctly. 6) Line 115: Double equal sign. Replace "==" with "=". 7) Line 159: "internal stress of soil" instead of "internal tress of soil"

---

## Author Comment (AC1) · 26 Oct 2020

In light of the two anonymous reviews, please find below our responses to the raised issues. First, we would like to gratefully thank the reviewers for their work in reading and reviewing the manuscript. Please know that all the proposed changes have been made to the manuscript.

In response to anonymous referee #1.

We thank the reviewer for this time in reading the manuscript. We have rewritten a large part of the introduction to clarify the scope and the potential application of this work. Now, the phenomena is first described, using terminology more commonly used within the literature. Afterwards, a short description of existing modelling approaches

and their shortcomings is provided. Finally, the introduction ends with the objective of the research: development of a new generalized semi-structured mass movement model.

In terms of the nature of the movements, we have clarified that the model implements structured movements (dynamics of a coherent mass), but similarly can (if required, or if the underlying physics indicates it) simulates fragmentation of the material. We have addressed our usage of the term "debris-flow" in our work. Instead we use "mass movement", as it more accurately reflects the generalized nature of the equations. Similarly to the work of Pudasaini (2012) and George and Iverson (2014) and Aaron and Hungr (2016), generalized sets of equations which are sometimes referred to as "debris flow" equations allow for simulation of a much wider range of phenomena.

The applicability of the model to granular flow is, when cohesive strength is insignificant, at least as good as the generalized two-phase equations from Pudasaini (2012) which is the predominant underpinning of this work. The influence of the additional work on cohesive strength and fragmentation has been developed with general validity in mind. When fragmentation occurs in the model, further runout reduces to the two-phase equations of Pudasaini automatically. However, full validation of the model to runout of various types of cohesive matrices must be further investigated. Finally, all specific comments have been addressed based on the reviewer suggestion.

In response to anonymous referee #2.

We thank the reviewer for this time in reading the manuscript. All the specific comments provided by the reviewer have been addressed in the manuscript. The sections have been re-labeled to be consistent and in line with the comments. Also, we have addressed our usage of the term debris-flow in this work. As with reviewer 1, we agree that mass movement (to be more generic) and specifically rock avalanches and landslide are more closely related to the applicability of this work.

References: Aaron, J., & Hungr, O. (2016). Dynamic simulation of the motion of

partially-coherent landslides. Engineering Geology, 205, 1-11. George, D. L., & Iverson, R. M. (2014). A depth-averaged debris-flow model that includes the effects of evolving dilatancy. II. Numerical predictions and experimental tests. Proceedings of the Royal Society A: Mathematical, Physical and Engineering Sciences, 470(2170), 20130820. Pudasaini, S. P. (2012). A general two‐phase debris flow model. Journal of Geophysical Research: Earth Surface, 117(F3).

---

## Author Response (AR2)

In light of the anonymous reviews, please find below our responses to the raised issues.

First, we would like to gratefully thank the reviewers for their work in reading and reviewing the manuscript. Please know that all the proposed changes have been made to the manuscript.

In response to anonymous referee #1.

We have adjusted and clarified issues in the introduction related the description of the process and the terminology related to the initiation.

[revised manuscript text omitted]